# Structural basis for MTA1c-mediated DNA N6-adenine methylation

Jiyun Chen [1,3], Rong Hu[1,3], Ying Chen[1,3], Xiaofeng Lin[1,3], Wenwen Xiang[1], Hong Chen[1], Canglin Yao[2] & Liang Liu [1 ✉]

DNA N6-adenine methylation (6 mA) has recently been found to play a crucial role in epigenetic regulation in eukaryotes. MTA1c, a newly discovered 6 mA methyltransferase complex in ciliates, is composed of MTA1, MTA9, p1 and p2 subunits and specifically methylates ApT dinucleotides, yet its mechanism of action remains unknown. Here, we report the structures of *Tetrahymena thermophila* MTA1 (TthMTA1), *Paramecium tetraurelia* MTA9 (PteMTA9)-TthMTA1 binary complex, as well as the structures of TthMTA1-p1-p2 and TthMTA1-p2 complexes in apo, S-adenosyl methionine-bound and S-adenosyl homo-cysteine-bound states. We show that MTA1 is the catalytically active subunit, p1 and p2 are involved in the formation of substrate DNA-binding channel, and MTA9 plays a structural role in the stabilization of substrate binding. We identify that MTA1 is a cofactor-dependent catalytically active subunit, which exhibits stable SAM-binding activity only after assembly with p2. Our structures and corresponding functional studies provide a more detailed mechanistic understanding of 6 mA methylation.

[1] State Key Laboratory of Cellular Stress Biology, School of Life Sciences, Xiamen University, Xiamen 361102 Fujian, China. [2] Department of Biological Sciences, University of California, San Diego, La Jolla, CA 92093, USA. [3] These authors contributed equally: Jiyun Chen, Rong Hu, Ying Chen, Xiaofeng Lin. ✉email: liangliu2019@xmu.edu.cn

DNA methylation is an important epigenetic modification that plays crucial roles in a variety of cellular processes, such as gene regulation, animal development, X-chromosome inactivation, and chromatin organization[1–4]. Methylation of N6-adenine (6 mA) on DNA was recently discovered to be present in diverse eukaryotes, though it exists at low levels in most of these organisms[5–10]. Unlike the well-studied C5-cytosine methylation (5 mC), the biological functions of 6 mA in eukaryotic genomic DNA is complicated and the various epigenetic mechanisms related to 6 mA remain to be further investigated[11]. Recently, the functional investigation of DNA methylation has revealed that the 6 mA is involved in transcription activation or repression, transgenerational epigenetic inheritance, nucleosome positioning, stress response, embryonic development, and tumor formation[12–17]. This suggests that 6 mA may function as a specific epigenetic mark in eukaryotes.

Writers, readers, and erasers are three distinct classes of enzymes, which are responsible for the regulation of 6 mA modification[18]. Readers and erasers participate in the recognition and removing of 6 mA, respectively. Writers are the DNA methyltransferases (MTases) that transfer a methyl group from S-adenosyl methionine (SAM) to the N6 position of an adenine residue to form 6 mA. In prokaryotes, multiple DNA adenine methyltransferases, such as CcrM, M.EcoP15I, M.TaqI, CamA and Dam, have been well studied[19–23]. M.TaqI, CamA and Dam are active as monomers, while CcrM and M.EcoP15I function as dimers, similar to eukaryote DNA cytosine methyltransferases Dnmt3a/3b/3L and DRM2, and RNA adenine methyltransferase METTL3-METTL14[24–27]. In contrast, the structure assembly and molecular basis of DNA 6 mA methyltransferases in eukaryotes are not fully understood.

Recently, studies on DNA 6 mA modifications in ciliates reported that the 6 mA not only directly disfavors nucleosome occupancy in a quantitative and sequence independent manner in vitro, but also impacts gene expression, cell growth, and sexual development in vivo[5,28,29]. The authors identified a 6 mA writer named MTA1c, which consisted of four ciliate proteins-termed MTA1, MTA9, p1 and p2. It is found that the four components of MTA1c are necessary for 6 mA methylation[5]. However, it remains unclear how MTA1c recognizes its DNA substrate specifically, and what is the molecular mechanism for four members of MTA1c working cooperatively to mediate 6 mA methylation.

In this study, we investigated the structural principles underlying the active mechanism of MTA1c in catalyzing 6 mA formation. We solved the crystal structures of MTA1-p2 binary complex and MTA1-p1-p2 ternary complex in SAM-bound, S-adenosyl homocysteine (SAH)-bound and SAM- or SAH-free states, as well as the structures of MTA1 and MTA1-MTA9 binary complex. Our structures together with biochemical studies revealed the structure assembly of MTA1c, as well as the specific role of each subunit in MTA1c. Our findings uncovered the details of MTA1c-mediated DNA N6-adenine methylation, providing the molecular basis for understanding the versatile functions of 6 mA.

## Results

### Assembly of the four-component MTA1c complex is driven by MTA1.
To explore the molecular mechanism of DNA N6-adenine methylation by the MTA1c complex, we first investigated the complex assembly of Tetrahymena thermophila (Tth) MTA1c using GST pull-down and size-exclusion chromatography (SEC) assays. GST-tagged p2 shows strong binding to MTA1, while no interactions were detected between GST-p2 with MTA9 and with p1 (Fig. 1a). Similarly, both GST-p1 and GST-MTA9 show strong binding affinity only for MTA1, and have no binding affinity

for other subunits. These data suggest that MTA1 specifically recognizes MTA9, p1 and p2 respectively. This conclusion is further confirmed by our GST pull-down assays with GST-tagged MTA1 and SEC assays using purified MTA1, MTA9, p1 and p2 proteins. MTA1 binds to one, two or three other subunits to form stable binary, ternary, or quaternary complexes, respectively (Fig. 1a, b and Supplementary Fig. 1a). In contrast, in the absence of MTA1, no interactions were detected between p1, p2 and MTA9 subunits (Fig. 1a). These data demonstrate that MTA1 specifically recognizes p1, p2 and MTA9 subunits to assemble into a stable methyltransferase complex, indicating that only MTA1 acts as the core subunit to drive the assembly of the quaternary MTA1c complex.

To investigate whether the subunits assembly affects the enzymatic activity of MTA1c, we then co-purified multiple stable complexes formed by MTA1 and other one, two or three subunits, and tested the activities of these complexes to catalyze 6 mA methylation of DNA in vitro (Supplementary Fig. 1b). MTA1, MTA1-MTA9 binary complex and MTA1-MTA9-p1 ternary complex almost show no activities to methylate DNA, while two ternary complexes MTA1-MTA9-p2 and MTA1-p1-p2 exhibit weak activities to methylate substrate, suggesting that p2 probably plays a particular role in mediating the activity of DNA methylation (Fig. 1c). Notably, the robust activity was observed when using the MTA1-MTA9-p1-p2 quaternary complex to catalyze oligonucleotide and long dsDNA 6 mA methylation, revealing that the MTA1c holoenzyme complex formation is necessary for 6 mA methylation (Fig. 1c and Supplementary Fig. 1c–e). In addition, the MTA1c complex shows no or weak detectable methylation activity for RNA and ssDNA, suggesting that it may specifically methylate dsDNA only (Supplementary Fig. 1f).

### Crystal structures of MTA1-p1-p2 ternary complex and MTA1-p2 binary complex.
To elucidate the detailed structural mechanism of MTA1c-mediated methylation of DNA N6-adenine, we determined the 2.7 Å resolution crystal structure of N-terminal truncated TthMTA1 (residues 126–372), in complex with Tthp1 and Tthp2, by the single wavelength anomalous diffraction (SAD) method and using a selenomethionine-labeled sample of the complex (Fig. 2a, b, Supplementary Fig. 2a and Supplementary Table 1).

The architecture of MTA1-p1-p2 ternary complex resembles a "wings-up" butterfly (Fig. 2b and Supplementary Fig. 2a). The NTD domain of p2 likes the "body", the MTase domain of MTA1 locating at one side of NTD constitutes the left "wing", the Helical domain of MTA1, HTH domain of p1, DBD and CTD domains of p2 positioning on the other side assemble into the right "wing". MTA1 is composed of the poorly conserved N-terminal Helical domain, followed by the conserved C-terminal MTase domain. The Helical domain far away the MTase domain contains a long α helix, which functions as an "arm" to grab p1 and p2 (Fig. 2b). The MTase domain consists of five α helices and one seven-stranded β sheet, forming a sandwich-shaped configuration (Supplementary Fig. 2b). p2 is made up of a predicted DNA binding domain (DBD) flanked by an N-terminal domain (NTD) and a C-terminal domain (CTD), each of them consisting of a long α helix and a short α helix (Supplementary Fig. 2c). As p1 was degraded by a putative protease during crystallization, in the determined MTA1-p1-p2 ternary complex structure, we only observed a single helix-turn-helix domain (HTH) consisting of two α helices connected by a turn region (Supplementary Fig. 2d).

To clarify how MTA1 assembles with p2 prior to p1 binding, we also determined the crystal structure of N-terminal truncated TthMTA1 (residues 126-372) in complex with Tthp2

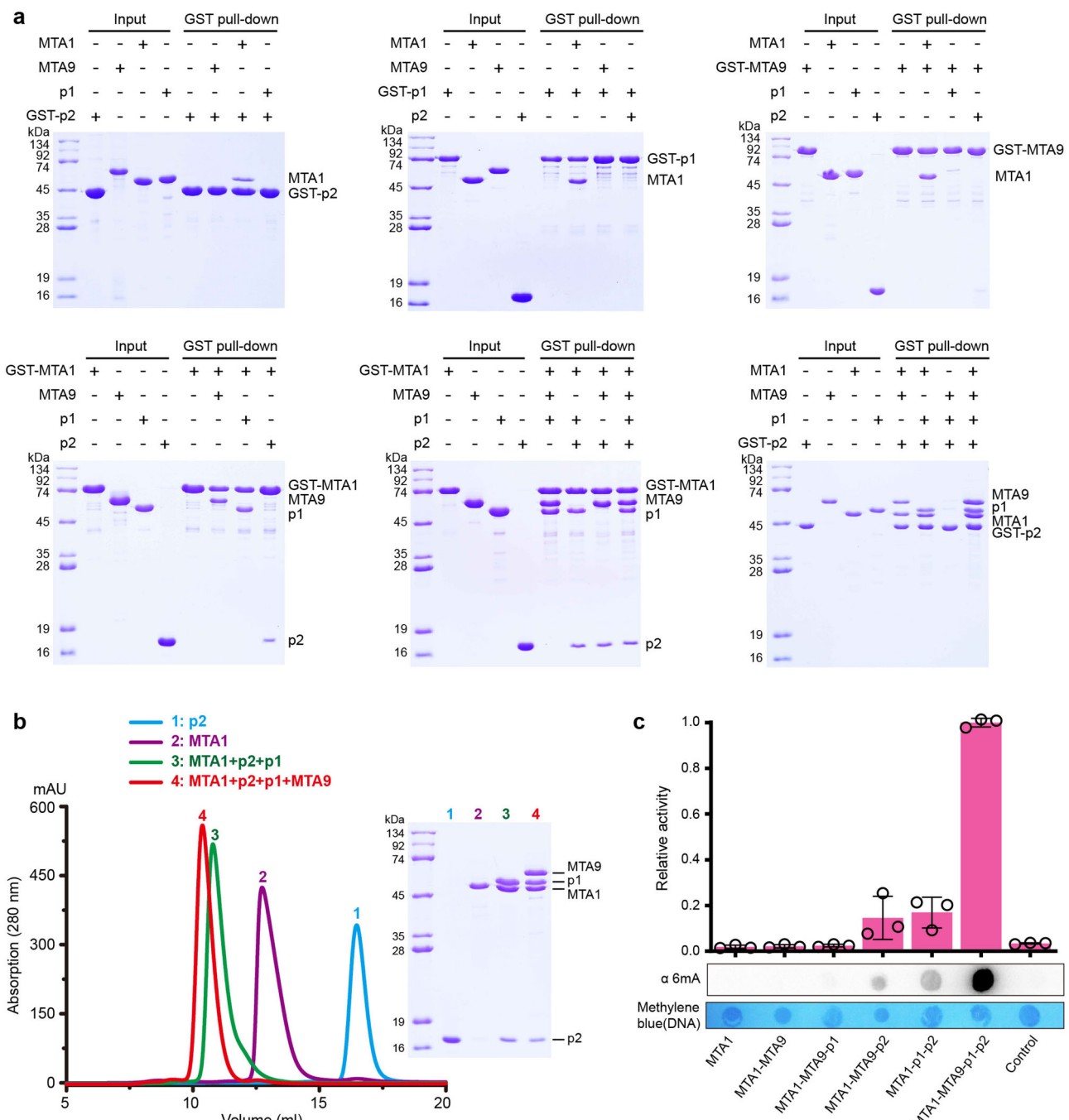

**Fig. 1 Identification of the subunit assembly of the TthMTA1c complex. a** GST pull-down assays for analyzing the direct interactions between MTA1, MTA9, p1 and p2. MTA1, MTA9, p1 and p2 were expressed in *Escherichia coli* and purified by Ni²⁺-agarose affinity column chromatography followed by size-exclusion chromatography. The SDS-PAGE gel was stained by Coomassie blue. **b** Gel filtration chromatography shows that MTA1 interacts with p1, p2 and MTA9. All samples were fractionated on a Superdex 200 increase 10/300, and the peak fractions were analyzed by SDS-PAGE. **c** Antibody-based methyltransferase activity of MTA1c complex and its subcomplexes on a 954-bp dsDNA. The dsDNA substrate was amplified by PCR from *Tetrahymena thermophila* strain SB210 genomic DNA. Data are shown as mean ± SD from *n* = 3 independent experiments; open circles indicate values for individual repeat measurements. Control, in the absence of MTA1c. Source data for (**a–c**) are provided in a Source Data file.

at 3.1 Å resolution (Fig. 2c and Supplementary Table 2). Our comparison between MTA1-p2 binary complex and MTA1-p1-p2 ternary complex structures reveals that MTA1 undergoes a modest conformational change upon p1 binding (Fig. 2d). Specifically, the N-terminal region of Helical domain is not visible in the MTA1-p2 structure, presumably due to the flexibility of this region, while it well ordered in the MTA1-p1-p2 structure (Supplementary Fig. 2e). This observation indicates

that p1 is able to stabilize the N-terminal region of the Helical domain of MTA1 upon binding to MTA1. Upon p1 binding, the Helical domain rotates toward the MTase domain, resulting in a narrower channel between the two domains, which is further stabilized by the interaction between p2 and MTA1 (Fig. 2b and Supplementary Fig. 2e). This channel is stabilized by MTA1, p1 and p2, revealing that it may be used to bind the substrate DNA. In addition, only minimal conformational changes of p2

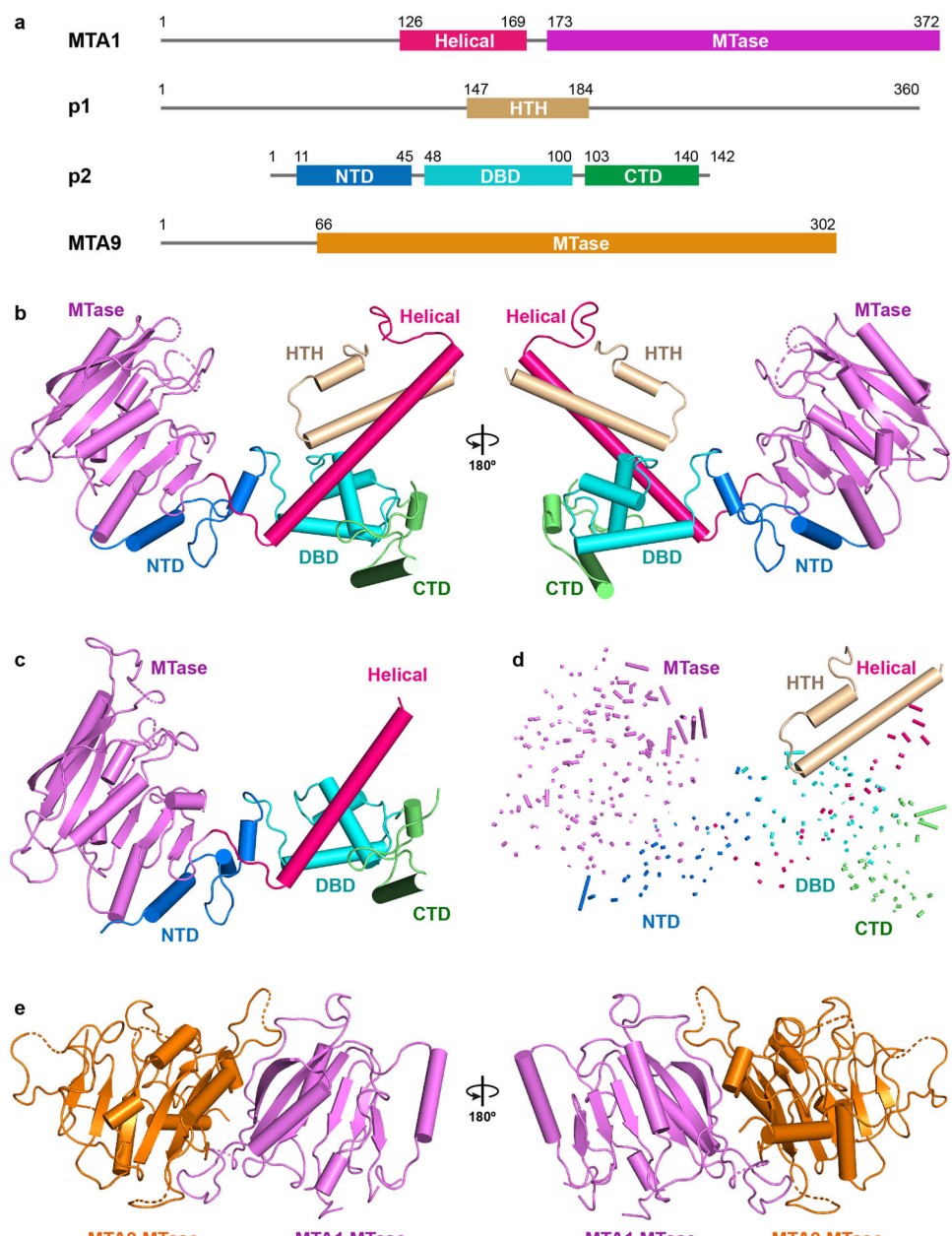

**Fig. 2 Structural overview of TthMTA1-Tthp1-Tthp2, TthMTA1-Tthp2 and TthMTA1-PteMTA9 complexes. a** Color-coded domain architecture of MTA1, p1, p2 and MTA9. **b** Front and rear views of the overall structure of the MTA1-p1-p2 ternary complex in ribbon representation. Domains of MTA1, p1 and p2 are colored according to the scheme used in (**a**). **c** Overall structure of the MTA1-p2 binary complex in ribbon representation. **d** Structural comparison of domain movement on proceeding from the MTA1-p2 binary complex to the MTA1-p1-p2 ternary complex. The Cα-Cα vector map is shown as lines. The vector lengths proportionally represent the domain movement between the two complexes being compared. **e** Front and rear views of the overall structure of the TthMTA1-PteMTA9 binary complex in ribbon representation.

are observed in these two complexes structures (Supplementary Fig. 2f), suggesting that p2 is obviously not involved in the recognition of p1.

**Structure of MTA1 in complex with MTA9.** Despite many attempts, the TthMTA9-TthMTA1 complex crystal was not obtained. Previous study has shown that 6 mA is abundant in *Paramecium tetraurelia* (Pte) and *Tetrahymena thermophila* (6 mA/A > 0.1%)[28]. Sequence alignment result shows that the MTA1c subunits are highly conserved between the two species. Both pull-down and co-purification assays demonstrate that TthMTA1 and PteMTA9 can form a stable complex, suggesting that interactions between MTA1c subunits in *Paramecium*

*tetraurelia* and *Tetrahymena thermophila* are probably conserved. To investigate MTA9 function on 6 mA methylation, we determined the crystal structure of PteMTA9 MTase domain (residues 66-302), in complex with TthMTA1 MTase domain (residues 171–372) at 3.1 Å resolution (Fig. 2e). In the structure of the MTA1-MTA9 complex, one MTA1 MTase binds to one MTA9 MTase, which is in agreement with our SEC data that the MTA1-MTA9 complex exhibits a 1:1 stoichiometry in solution. The MTase domain of MTA9 consists of six α helices and one six-stranded β sheet (Supplementary Fig. 2g), forming a sandwich-shaped configuration, similar to MTA1 MTase. The MTA1 and the MTA9 form a stable antiparallel heterodimer in an asymmetric way through extensive contact. The overall

structure of this complex also adopts a similar "wings-down" butterfly appearance observed for the METTL3-METTL14 heterodimer complex[30–32].

Notably, in order to reveal the structural assembly of the MTA1c complex, we used some truncations or combinations for crystallization and structural determination. Truncation of TthMTA1 (residues 126–372), TthMTA9 (residues 67–449) and Tthp1 (residues 1–309) has no significant effect on enzyme activity of MTA1c (Supplementary Fig. 2h).

**Specific recognition of p1 and p2 by MTA1**. MTA1 contains the MTase and Helical domains, forming a central channel that is responsible for specific recognition of p1 and p2 (Supplementary Fig. 3a, b). The MTase domain specifically recognizes the N-terminal α helices of p2 (Fig. 3a). The side chains of His198, Gln195, Asp174 and Arg358 within the MTase domain form hydrogen bonds with the side chains of Asn20, Asp24, Tyr17 and His39, as well as the main chains of Tyr41 and Thr44 within the α1 and α2 helices of p2. In addition, the main chains of Asp174, Ser176, Pro178, Cys180 and Leu372 within MTase domain are involved in a hydrogen-bonding network with the side chains of Arg22, Ser26, Asn27 and Tyr41 in the NTD domain of p2. The C-terminal region of the Helical domain of MTA1 mainly recognizes the DBD and CTD domains of p2. There are extensive hydrogen bonds and hydrophobic interactions between these domains, playing a crucial role in stabilizing MTA1-p2 architecture. We generated multiple alanine mutations or truncations of MTA1 or p2 residues involved in intermolecular contacts and investigated the impact of these mutations and truncations on the binding of MTA1 and p2. Either alanine substitutions or truncations of residues in p2 resulted in reduced or complete loss in binding affinity to MTA1 (Fig. 3b). Similar effects were observed with the mutant and N-terminal truncated proteins of MTA1, which almost completely lost the ability to bind to p2 (Fig. 3c and Supplementary Fig. 3c).

In addition, the N-terminal region of the Helical domain of MTA1 participates in recognition of p1 (Fig. 3a). The side chains of Leu139, Leu142, Ile146, IIe150 and Tyr153 within the Helical domain of MTA1 interact with the side chains of Leu181, Ile177, Leu173, Leu170 and Leu167 in HTH domain of p1, forming multiple hydrophobic interactions, as is seen in the leucine zipper proteins. The p1 alanine mutants L167A, L170A, L173A and I177A all show reduced binding affinity to MTA1, with the double mutant L167A/L170A, triple mutant L167A/L170A/L173A and quadruple mutant L167A/L170A/L173A/I177A, resulting in essentially complete loss in binding affinity (Fig. 3d). Notably, the MTA1 mutants also has no or relatively weak ability to bind to p1 wild-type protein (Supplementary Fig. 3d). Interestingly, the interaction residues are highly conserved in Helical domain of MTA1 and HTH domain of p1, despite the fact that both domains have very low sequence identity (Fig. 3e), indicating that these hydrophobic amino acids are critical for forming the hydrophobic interactions between MTA1 and p1. With multiple the N-terminal or the C-terminal truncated proteins of p1, we further confirmed that MTA1 directly interacts with HTH domain of p1 using GST pull-down assays. The HTH domain retained GST-p1 (residues 1-185) and GST-p1 (residues 146-360) proteins show strong ability to bind to MTA1, whereas the HTH domain deleted GST-p1 (residues 1-145) and GST-p1 (residues 186-360) proteins completely lose their binding affinity (Supplementary Fig. 3e).

The MTA1 is able to form stable complex with p1 and p2, primarily because it has a unique Helical domain. Structure and sequence alignments revealed that other structure-known DNA or RNA MTases, such as MTA9, METTL3 and DNMT1, all lack

the Helical domain, though they possess a similar sandwich-shaped MTase domain (Supplementary Fig. 3f and Supplementary Fig. 4)[30,31,33]. Though an N-terminal α-helical motif (NHM) is identified in RNA MTase METTL14, it shows no structural similarity and sequence homology with the Helical domain of MTA1. It suggests that the MTA1 has structural characteristics that are obviously different from other nucleic acid MTases, and these structural characteristics probably associate with its specific function.

**Specific recognition of MTA9 by MTA1**. The MTase domain of MTA1 specifically recognizes the MTase domain of MTA9 through multiple hydrogen bonds and hydrophobic interactions (Fig. 3f). MTA1 and MTA9 each has an interface loop, playing a crucial role in complex stabilization. The residues of Tyr294, Leu295, Gln296 and His297 within the interface loop (residues 284–298) of MTA1 interact with the residues of Arg231, Phe142, Leu168, Leu275, Arg246, Thr247, Pro248 and Glu166 of MTA9, forming multiple hydrogen bonds and hydrophobic interactions (Fig. 3f and Supplementary Fig. 3g). The residues of Lys218, Val219, Leu220, Asn221 and Gln222 in the interface loop (residues 218–224) of MTA9 are recognized by the residues of MTA1 through specific interactions. In addition, the side chain of Tyr258 in MTA1 forms stacking interaction with the side chain of Tyr164 in MTA9, further stabilizing the MTA1-MTA9 complex (Supplementary Fig. 3h). Mutation studies reveal that interface loop mutants of MTA1 and MTA9, and Y164A mutant of MTA9 disrupt the association between MTA1 and MTA9 (Fig. 3g, h), indicating that the interface loops and the residue Tyr164 of MTA9 play critical roles in the formation of the stable MTA1-MTA9 complex.

**Recognition the methyl donor SAM by MTA1-p1-p2 complex**. In order to investigate MTA1-mediated key catalytic process, we determined crystal structures of the SAM donor-bound MTA1-p1-p2 complex and the SAH product-bound MTA1-p1-p2 complex by soaking approach at 3.4 Å and 3.0 Å resolution, respectively (Fig. 4a, b). In both structures, clear electron densities for the SAM and SAH are only observed in MTA1 (Supplementary Fig. 5a, b), no ligand electron densities are observed in p1 and p2. Both SAM and SAH molecules are positioned in a narrow pocket formed by multiple loops at the ends of β1, β2, β5, β6 and β7, showing very similar conformations.

Structure analysis reveals that SAM is stabilized in the pocket by extensive hydrogen bonds formed between MTA1 and SAM (Fig. 4a, b). The methionine moiety of SAM is surrounded by the side chains of Asp209, Lys334, Arg357, Asn359 and Asn360, while the hydroxyl groups of ribose are hydrogen bonded to Glu371. The adenine moiety is coordinated by hydrogen binding interactions to the side chains of Asp182 and Asn370, as well as the main chain of Val183. In addition, Asp209, Glu353 and Phe355 form water-mediated interactions with SAM. The SAM molecule is primarily coordinated by eleven residues, of which Asp182, Asp209 and Asn360 are highly conserved in MTA1 and METTL3 MTases (Supplementary Fig. 4). Alanine substitution of each residue of them not only abolishes the MTase activity (Fig. 4c), but also dramatically reduces binding affinity to SAM (Fig. 4d), indicating that these three amino acids are critical for SAM binding. Moreover, the R357A, N359A and N370A mutants show significantly reduced enzyme activity (Fig. 4c), but still maintain a strong SAM-binding affinity (Supplementary Fig. 5c).

**p2 directly activates the cofactor-binding activity of MTA1**. We tested the binding ability of TthMTA1 and TthMTA9 to SAM by isothermal titration calorimetry (ITC). Unexpectedly, neither the

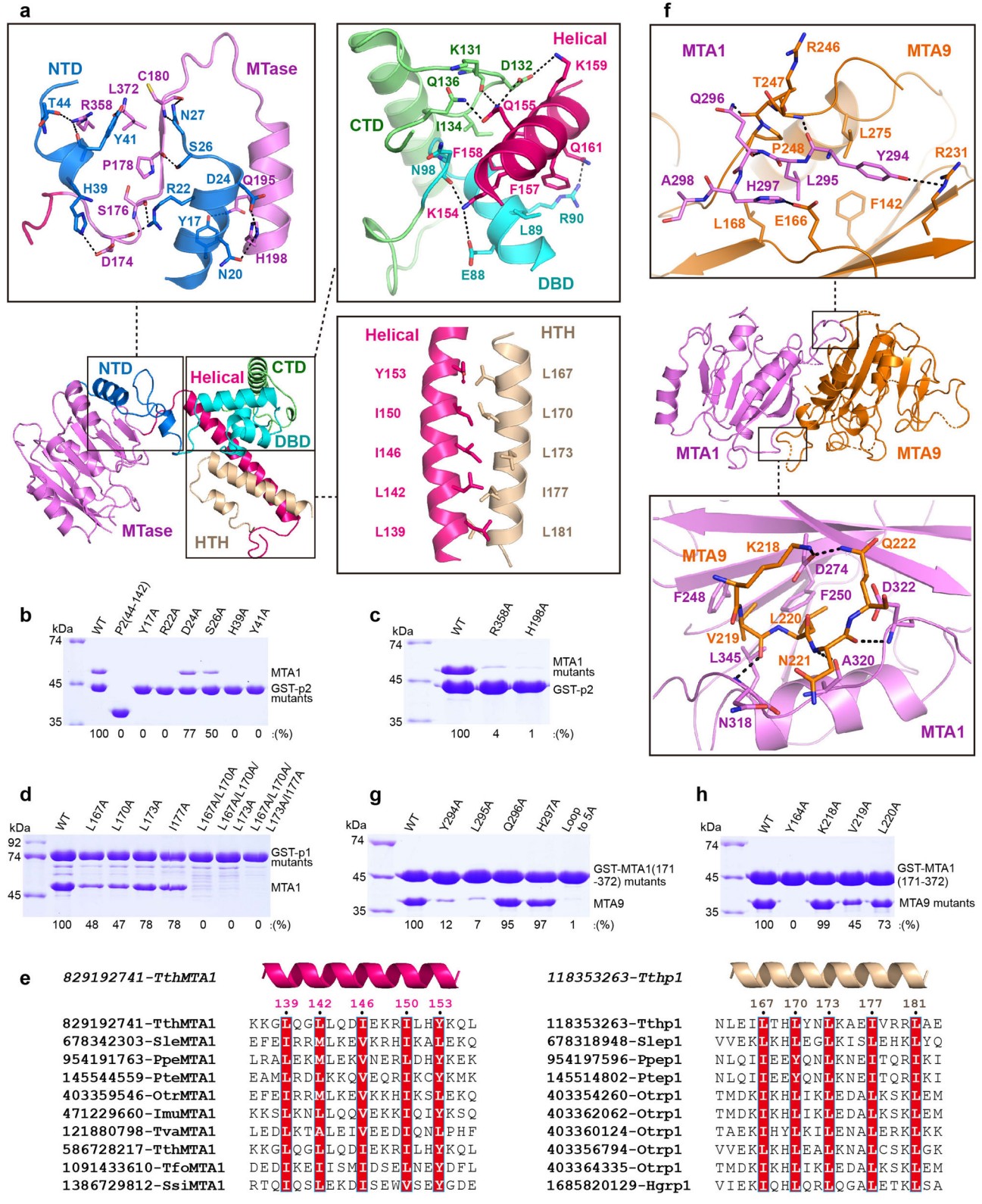

MTA1 nor the MTA9 single subunit protein has detectable SAM-binding activity (Fig. 5a). As a SAM molecule is clearly observed in the MTA1 pocket of the MTA1-p1-p2 ternary complex structure, we speculated p1 or p2 should play an essential role in SAM binding for MTA1. We then investigated this hypothesis with TthMTA1c complex, and multiple combinations of two- or three-subunit subcomplexes. Interestingly, we found that these complexes show detectable binding affinities to SAM, only when

in the presence of MTA1 and p2 both subunits (Fig. 5a). These observations suggest that the cofactor substrate binding activity of MTA1 is certainly activated by p2. This suggestion is further confirmed by our SAM- and SAH-bound MTA1-p2 structures. Two ligands bind stably in the narrow pocket of MTA1 subunit as observed in the ligands-bound MTA1-p1-p2 structures (Supplementary Fig. 5d, e). Structure analysis of SAM-bound MTA1-p2 complex shows that two loops at the end of β6 and β7 responsible

**Fig. 3 Structural details of TthMTA1-Tthp1-Tthp2 and TthMTA1-PteMTA9 intermolecular contacts. a** Detailed interactions between MTA1 and p2, as well as between MTA1 and p1 in the MTA1-p1-p2 complex. **b** GST pull-down experiments assaying the ability of GST-p2 mutants to interact with MTA1. MTA1 proteins were incubated with wild-type or mutant GST-p2 proteins immobilized on glutathione resin. The bound MTA1 proteins were quantitatively analyzed by the gray scanning. **c** GST pull-down experiments assaying the ability of MTA1 mutants to interact with GST-P2. **d** GST pull-down experiments assaying the ability of GST-p1 mutants to interact with MTA1. **e** Sequence alignments of hydrophobic α helices in the MTA1 (left panel) and p1 (right panel) proteins. Each sequence is labeled with its GenBank Identifier (GI) number and the systematic name of an organism. The conserved hydrophobic residues are shaded in red. **f** Detailed interactions between MTA1 and MTA9 in the MTA1-MTA9 complex. **g** GST pull-down experiments assaying the ability of GST-MTA1 (residues 171–372) mutants to interact with MTA9. **h** GST pull-down experiments assaying the ability of MTA9 mutants to interact with GST-MTA1 (residues 171-372). Source data for (**b–d**, **g–h**) are provided in a Source Data file.

for SAM binding pocket formation, are stabilized by p2 subunit (Fig. 5b). The side chain of Arg358 within one loop (Loop1) of MTA1 interacts with the main chains of Thr44 and Tyr41 in NTD domain of p2, while the main chain of Leu372 within the other loop (Loop2) of MTA1 forms hydrogen bond with the side chain of Tyr41 in NTD domain of p2. Notably, residues of Arg357 and Asn359 adjacent to Arg358, as well as residue of Glu371 linked to Leu372, are involved in SAM recognition. R358A of MTA1 and Y41A of p2 mutations both completely abolished SAM-binding activity of MTA1 (Fig. 5c), indicating that these hydrogen bonds interactions between MTA1 and p2 are essential for SAM binding.

TthMTA1 can form stable complex with both Tthp2 and Ptep2, suggesting that interactions between MTA1 and p2 subunits in *Tetrahymena thermophila* and *Paramecium tetraurelia* are likely conserved. To confirm whether Ptep2 can also activate the cofactor-binding activity of TthMTA1, we tested the binding ability of TthMTA1 to SAM by ITC in the presence of Ptep2, Ptep1 or PteMTA9. TthMTA1 shows detectable binding affinity to SAM in the presence of Ptep2, while it exhibits no detectable SAM-binding activity in the presence of Ptep1 or PteMTA9 (Fig. 5d). This observation indicates that Ptep2 can function as Tthp2 to activate TthMTA1 for SAM binding. Then we determined crystal structure of the SAM donor-bound TthMTA1-Ptep2 complex (Fig. 5e). As observed in the SAM-bound TthMTA1-Tthp2 complex structure, SAM molecule binds stably in the narrow pocket of MTA1. In the SAM-binding pocket, Loop1 and Loop2 are stabilized by the NTD domain of p2 and participate in SAM recognition. These structural and binding data further confirm that p2 plays an activating function for MTA1 binding SAM molecule.

**Induced-fit mechanism for SAM-binding**. The MTase domain is highly conserved in both MTA1 and METTL3, especially the catalytic site (DPPW motif) and the key residues responsible for SAM binding, are nearly identical (Supplementary Fig. 6a)[30,31]. In the absence of METTL14, the single METTL3 subunit also shows a high binding affinity to SAM (Supplementary Fig. 5f)[34]. However, we didn't detect any binding affinity between SAM and the MTA1 regardless of the presence or absence of MTA9. We compared the SAM-free MTA1 structures in the MTA1-MTA9, MTA1-p2 and MTA1-p1-p2 complexes, and observed significant conformational differences in the SAM-binding pocket (Fig. 6a). In the MTA1-MTA9 structure, the side chains of Arg357 and Asn370 are adjacent to the side chain of the catalytic residue Pro211, occupying the binding position of SAM (Fig. 6b). While in the MTA1-p2 and MTA1-p1-p2 structures, the side chains of Arg357 and Asn370 are far away from the catalytic residue Pro211 and have a wide binding pocket for the cofactor substrate. Structural superposition of the MTA1-MTA9 complex with SAM-bound MTA1-p2 complex reveals that the significant conformational changes of the SAM-binding pocket are likely resulted by p2 binding. The main chains of Tyr41 and Thr44 of p2 bind to the side chain of Arg358, inducing Loop1 of MTA1 moves away from the catalytic residue Pro211. While the side

chain of Tyr41 of p2 interacts with the main chain of Leu372, resulting Loop2 of MTA1 shifts away from the catalytic motif (Fig. 6c). These conformational changes generate a wide pocket for MTA1 to bind to SAM (Fig. 6c). We reason that MTA1 exists in unsteady state, with a relatively closed SAM-binding pocket obstructing the complete accommodation of SAM prior to p2 binding, but switching to the steady state upon p2 binding, resulting in an open SAM binding pocket to perfectly fit to SAM.

To better understand the molecular mechanism for p2-mediated MTA1 activation, we also determined the structure of MTA1 MTase domain (residues 171–372) at 1.8 Å resolution (Supplementary Fig. 6b and Supplementary Table 3). However, in the crystal, the β7 strand extends toward the neighboring MTA1 molecule, possibly due to crystal packing, and forms a β-sheet with its β1–β6 strands, leading to the formation of SAM-binding pocket by two neighboring MTA1 molecules. The Loop1 located at the end of β6 is derived from one MTA1 molecule, whereas the Loop2 positioned at the end of β7 is derived from the other neighboring MTA1 molecule (Supplementary Fig. 6c, d). Structural comparison of the MTA1 to the MTA1-p2 complex demonstrates that the SAM-binding pocket exhibits distinct conformation (Supplementary Fig. 6e, f). In the MTA1 structure, the residues of Asn370 and Glu371 in the Loop2 from the neighboring MTA1 occupy the SAM-binding site, the side chain of Arg357 within the Loop1 forms hydrogen bond with the side chain of Asp209 in the catalytic center to block the binding of SAM. While in the structure of MTA1-p2 complex, the β7 strand, Loop1, and Loop2 of MTA1 are stabilized by the NTD domain of p2 (Supplementary Fig. 6e), thus generating an activated SAM-binding pocket (Supplementary Fig. 6f). These observations indicate that p2 is able to act as a molecular switch to regulate the SAM-binding activity of MTA1.

In addition, MTA1 and MTA9 share a classic α-β-α sandwich architecture (Fig. 6d). Structural superposition of the MTA1 with MTA9 shows that the MTA9 has a putative SAM-binding pocket near the catalytic center (Fig. 6e). Structural analysis reveals that the catalytic motif and almost all of the residues involved in SAM recognition in the MTA1 pocket are not conserved in the MTA9 pocket (Fig. 6f).

**MTA1, MTA9, p1, p2 cooperate to bind DNA**. To determine whether the four subunits of MTA1c are necessary for substrate recognition, we investigated the effect of each subunit on substrate DNA binding using electrophoretic mobility shift assays (EMSA). The MTA1c holoenzyme complex shows strong binding affinity with DNA substrate (Fig. 7a). However, the complex, lacking one or two subunits, exhibits significantly reduced binding affinity with DNA substrate. These results suggest that the four subunits collectively contribute to internal DNA binding. In addition, SAM binding does not improve the affinity of MTA1c for DNA and vice versa (Supplementary Fig. 7a, b).

Both GST pull-down assays and crystal structure reveal a heterodimeric association of MTA1 and MTA9, similar to METTL3 and METTL14 (Supplementary Fig. 7c). A positively

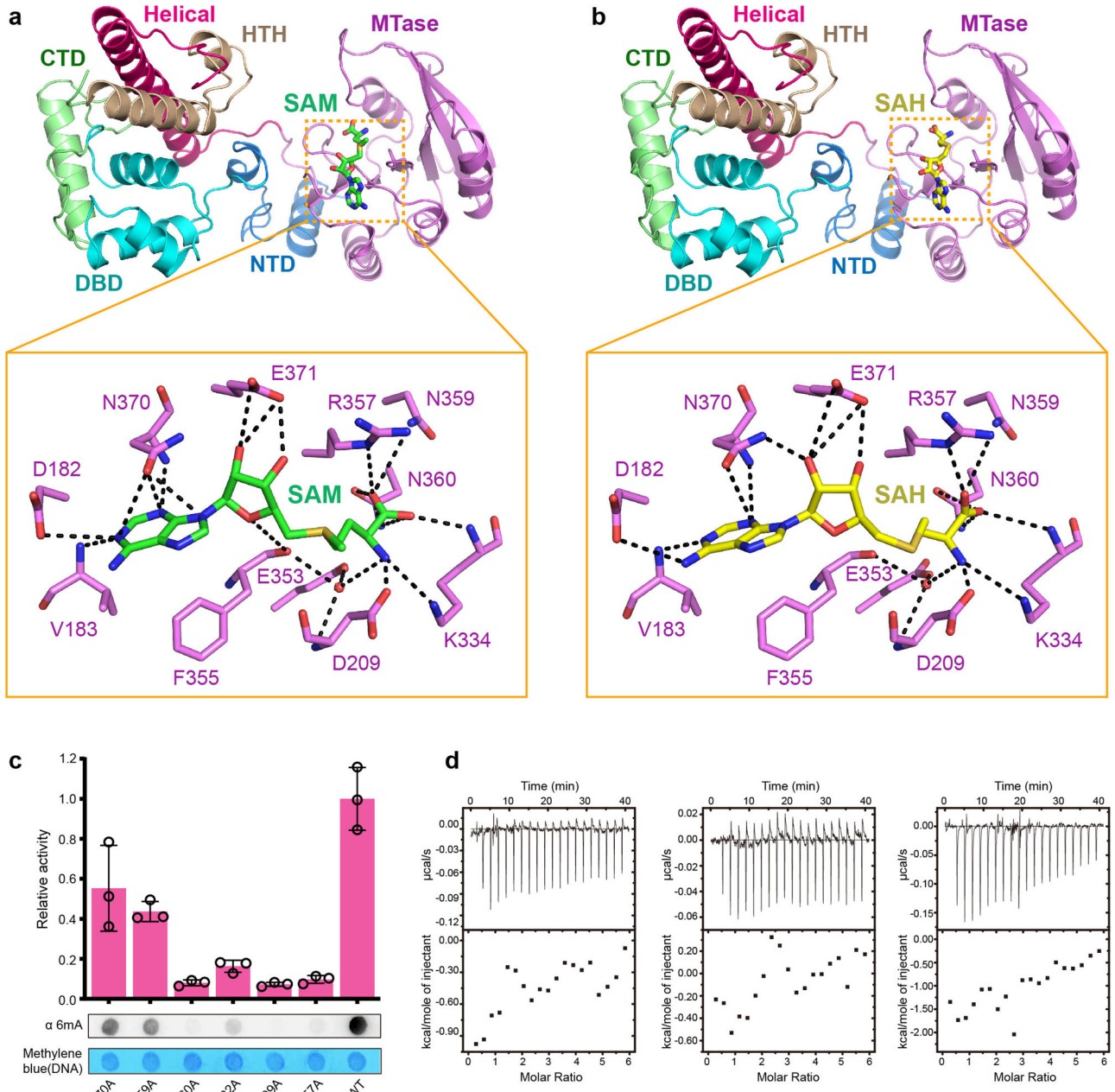

**Fig. 4 Recognition of the cofactor substrate. a** Detailed interactions between SAM and MTase domain of TthMTA1. Residues involved in SAM binding are shown as violet sticks. SAM is shown as green sticks. Water is shown as red spheres. Hydrogen bonds are shown as black dashed lines. **b** Detailed interactions between SAH and MTase domain of TthMTA1. Residues involved in SAH binding are shown as violet sticks. SAH is shown as yellow sticks. Water is shown as red spheres. Hydrogen bonds are shown as black dashed lines. **c** Antibody-based methyltransferase activity of the TthMTA1c complexes with indicated MTA1 point mutations. Each mutation does not affect MTA1-MTA9-p1-p2 complex formation. Each MTA1-MTA9-p1-p2 mutant complex used for enzymatic assay was expressed and purified as MTA1-MTA9-p1-p2 WT complex. A 954-bp dsDNA PCR product was sued as the substrate. Data are shown as mean ± SD from $n = 3$ independent experiments; open circles indicate values for individual repeat measurements. **d** ITC assay of the TthMTA1c complexes with indicated MTA1 point mutations. Each MTA1-MTA9-p1-p2 mutant complex used for ITC assay was purified as MTA1-MTA9-p1-p2 WT complex. Source data for (**c**) are provided in a Source Data file.

charged groove between METTL3 and METTL14 is likely responsible for RNA binding[30]. However, such groove is likely not to exist between MTA1 and MTA9 (Supplementary Fig. 7d). It is reported that two tandem CCCH-type zinc fingers located at the N-terminus of METTL3 enhance interaction with substrate RNA[31,34]. While the N-terminus of MTA1, lacking the CCCH zinc fingers, appears to be unnecessary for DNA recognition, because its truncation has no effect on methylation activity (Fig. 7b).

Based on the structural information, we built a structural model of MTA1c-DNA complex (Fig. 7c). In this model, MTA1c was constructed from the superimposition of MTA1-MTA9 structure and MTA1-p1-p2 structure, and DNA was docked by the structural superimposition of MTA1-p1-p2 complex with DNA-bound M.TaqI MTase. Structural comparison shows a structural similarity between the MTA1-p1-p2 complex and the M.TaqI MTase (Supplementary Fig. 7e)[23], revealing a possible DNA binding channel between two wings

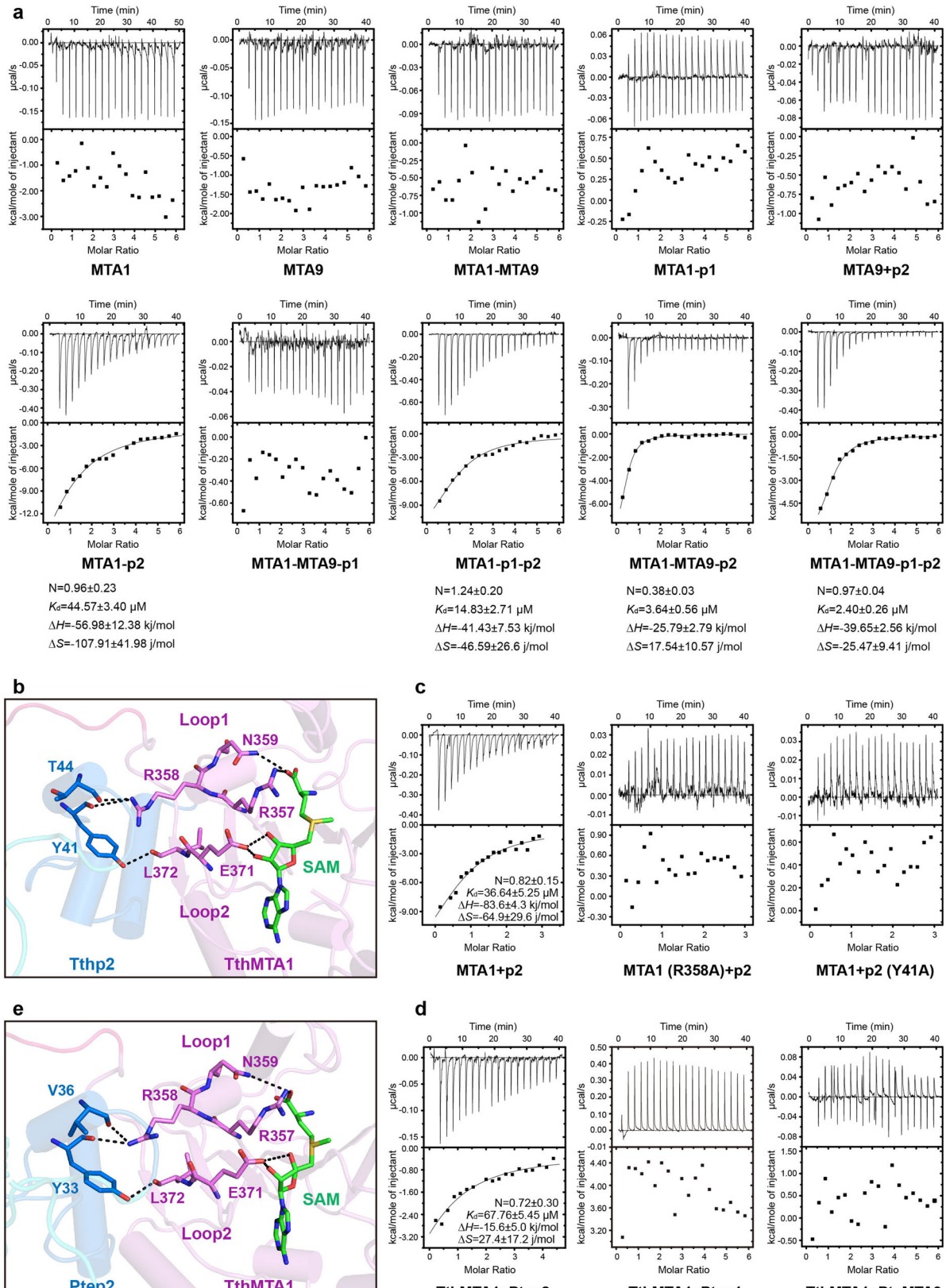

**Fig. 5 p2 induced the activation of MTA1. a** ITC assay using single TthMTA1, TthMTA9, as well as two-component, three-component, and four-component TthMTA1c complexes. **b** Detailed interactions between the SAM-binding loops of TthMTA1 and the NTD domain of Tthp2. **c** ITC assay measuring the SAM-binding affinity of wild-type or mutant TthMTA1 in the presence of wild-type or mutant Tthp2. **d** ITC assay measuring the SAM-binding affinity of TthMTA1 in the presence of Ptep2, Ptep1 or PteMTA9. **e** Detailed interactions between the SAM-binding loops of TthMTA1 and the NTD domain of Ptep2.

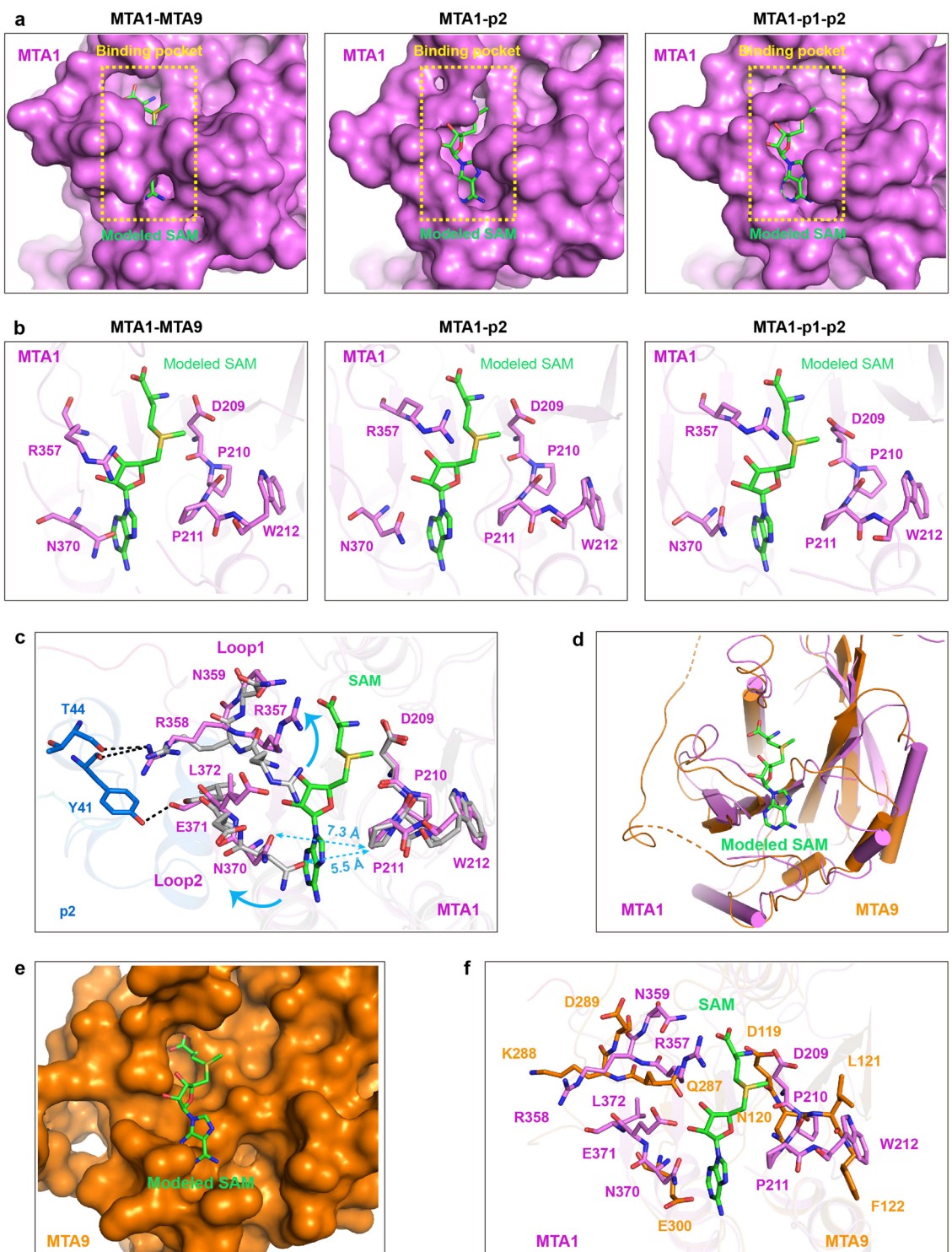

**Fig. 6 Conformational change of MTA1 upon p2 binding. a** Structural comparison of the SAM-binding pocket of MTA1 in the MTA1-MTA9 (left panel), MTA1-p2 (central panel) and MTA1-p1-p2 (right panel) complexes. Each MTA1 structure shows SAM-binding pocket in the SAM-free state. For comparison, a SAM molecule was modelled in the SAM-binding pocket of MTA1. **b** The SAM-binding pocket presents significantly different conformations in the structures of MTA1-MTA9 (left panel), MTA1-p2 (central panel) and MTA1-p1-p2 (right panel) complexes. Each MTA1 structure shows SAM-binding pocket in the SAM-free state. For comparison, a SAM molecule was modelled in the SAM-binding pocket of MTA1. **c** Superposition of the SAM-binding pocket within MTA1 in the MTA1-MTA9 complex (MTA1 in gray) with that in the SAM-bound MTA1-p2 (SAM in green, MTA1 in violet and p2 in marine) complex. **d** Structural comparison between MTA1 (violet) and MTA9 (orange). **e** MTA9 shows a potential SAM-binding pocket, a SAM molecule (green) was modelled in the potential SAM-binding pocket. **f** Superposition of the potential SAM-binding pocket of MTA9 (orange) with the SAM-binding pocket of MTA1 in the SAM-bound MTA1-p2 (SAM in green, MTA1 in violet) complex.

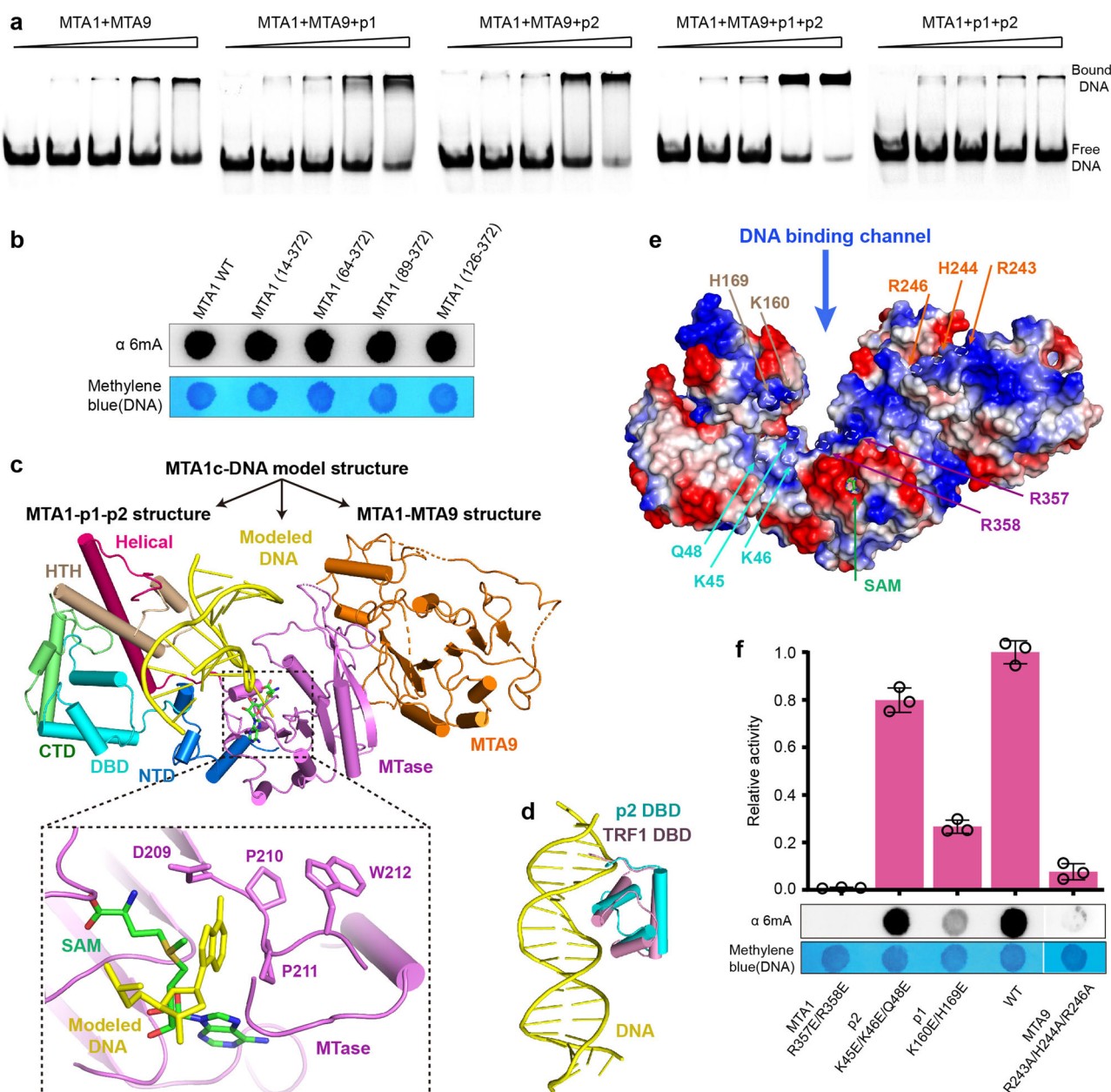

**Fig. 7 Substrate DNA binding by the MTA1c complex. a** Electrophoretic mobility shift assay using two-component, three-component, and four-component MTA1c complexes. **b** Antibody-based methyltransferase activity of the MTA1-MTA9-p1-p2 complexes with indicated MTA1 N-terminal truncations. A 954-bp dsDNA PCR product was sued as the substrate. **c** A structural model of the substrate dsDNA-bound MTA1c complex. The MTA1c complex structure was generated from the MTA1-MTA9 and MTA1-p1-p2 subcomplex structures. The dsDNA was modelled in the substrate DNA-binding site of MTA1c complex by superimposition of the M.TaqI-DNA complex structure (PDB: 1G38) onto the MTA1c complex. The modeled DNA is shown in yellow. **d** Superimposition of the TRF1 DBD-DNA complex structure (PDB: 1W0T) onto the DBD domain of p2. **e** The surface electrostatic potential calculated with PyMOL of MTA1c complex. **f** Antibody-based methyltransferase activity of the MTA1-MTA9-p1-p2 complexes with indicated TthMTA1, PteMTA9, Tthp1 or Tthp2 mutations. A 954-bp dsDNA PCR product was sued as the substrate. Data are shown as mean ± SD from $n = 3$ independent experiments; open circles indicate values for individual repeat measurements. Source data for (**a**, **b**, **f**) are provided in a Source Data file.

of the MTA1c complex. A dsDNA substrate was perfectly docked in the possible DNA binding channel in the MTA1c model structure (Fig. 7c). The MTase domain of MTA1 and the HTH domain of p1 bind to the major and minor grooves of this DNA, respectively. MTA9 may serve as a scaffold protein to further stabilize the binding between MTA1 and DNA (Supplementary Fig. 7f). The DBD domain of p2 located at the bottom of the channel is also likely to participate in the recognition of the DNA substrate, due to its high structural homology with the TRF1 and TRF2 DBD domains (Fig. 7d),

which are involved in the specifically recognition of the double-stranded telomeric DNA[35]. Surface electrostatic potential analysis of the modelled MTA1c complex exhibits that this putative DNA binding channel is highly positively charged, and has the potential to form extensive electrostatic contacts and hydrogen bonds with the DNA substrate (Fig. 7e). Substitutions of multiple positively charged residues in the channel abolish or reduce DNA binding and N6-adenine methylation (Fig. 7f and Supplementary Fig. 8), revealing that the positively charged channel is essential for DNA N6-adenine methylation.

## Discussion

Our results reveal that MTA1 recruits MTA9, p1 and p2 subunits to assemble into a stable heterotetramer, which is critical for substrate DNA recognition and subsequent N6-adenine methylation. Through structural and biochemical studies, we investigated the detailed division of labor between MTA1, MTA9, p1 and p2 subunits. MTA1 drives a heterotetrameric MTase assembly through the binding of its Helical domain to p1 and p2 and the interaction of its MTase domain with MTA9. MTA1 is the catalytically active subunit, but almost has no detectable SAM-binding affinity prior to assembly with p2. An important function of p2 is to activate the SAM-binding activity of MTA1 by opening the SAM-binding pocket of MTA1 through its N-terminal NTD domain. MTA1 contains a highly conserved catalytic motif (DPPW), which is well known in METTL3 and is involved in SAM recognition and methyl transfer. In contrast to MTA1, MTA9 has a potential catalytic site, which is not strictly conserved and has little effect on SAM-binding affinity and DNA methylation activity of MTA1c after mutation (Supplementary Fig. 9). The combination of MTA9 and MTA1 is probably able to stabilize the conformation of MTA1c complex and thus improve its catalytic activity. In addition, p1 and p2 subunits form a central positively charged channel with the MTase domain of MTA1, similar to the DNA binding channel of M.TaqI. Both p1 and p2 subunits are homeobox-like proteins, and both also contribute to DNA binding. In consequence, we speculate that p1 and p2 subunits play important roles in maintaining complex integrity and stabilizing substrate DNA binding. Taken together, MTA1, MTA9, p1 and p2 are interdependent in terms of assembly, stability and enzyme activity, so there is a high degree of synergy between the four essential subunits.

The accessory factor-dependent SAM recognition may be unique to MTA1. On the basis of our structural findings and biochemical data, we propose a specific activation mechanism by which MTA1 binds to SAM. Structure comparison between MTA1 and MTA1-p2 complex indicates that prior to p2 binding, MTA1 has a narrow SAM-binding pocket, which will definitely hinder the specific recognition of SAM and lead to a low binding affinity for SAM. We suspect that the active site of MTA1 is likely to exhibit an unsteady conformation before binding to p2. The side chains of Arg357 and Asn370 are very close to the side chains of catalytic residues, forming a closed SAM-binding pocket that almost has no or weak SAM-binding affinity. The binding of p2 induces a relatively broad SAM-binding pocket formation of MTA1 through the specific interactions between the NTD domain of p2 and the SAM-binding loops of MTA1, thus being able to fit SAM completely. The transition of MTA1 from incomplete fitting state to complete fitting state is mainly due to large conformational changes in SAM-binding pockets resulted by accessory factor, reminiscent of the m6A methyltransferase METTL16. The methylation activity of METTL16 can be regulated by a polypeptide loop near the SAM-binding site by hindering the SAM-binding pocket[36–38]. Notably, MTA1 and METTL16 have two distinct activity regulators, although both modulate activity through the SAM-binding pocket.

The shared trait of MTA1c and METTL3-METTL14 enzymes is that they possess SAM-dependent methyltransferase activity, with the ability to transfer a methyl group from SAM to nucleic acid substrate. However, their structures and catalytic mechanisms show significant difference. Structural comparison of MTA1, MTA9, METTL3, and METTL14 reveals that MTA1 possesses a unique Helical domain, which is responsible for the recognition of accessory factors p1 and p2. The Helical domain adopts as an "arm" conformation, playing a key role in the stabilization of MTA1-p1-p2 complex. In the absence of Helical domain, MTA9, METTL3 and METTL14 are probably unable to associate with

other accessory factors, such as p1 and p2, which may be essential for DNA binding. Two CCCH zinc fingers located at the N-terminus of METTL3 are required for RNA methyltransferase activity due to its RNA binding ability, while these fingers are not identified in MTA1[31]. These structural differences between MTA1c and METTL3-METTL14 complexes are likely to provide a molecular explanation for why MTA1c has DNA methylation activity but likely no RNA methylation activity, whereas METTL3-METTL14 has the opposite.

In addition, there are quite different substrate sequence specificity and substrate binding mode between MTA1c and METTL3-METTL14. MTA1c preferentially methylates ApT dinucleotides in dsDNA, while METTL3-METTL14 specifically recognizes the RRACH sequence in RNA for methylation. MTA1c complex forms a centrally positively charged channel that may be responsible for substrate DNA binding. Distinct from MTA1c, METTL3 and METTL14 interact with each other to generate a positively charged groove that is likely to be involved in substrate RNA recognition.

Lastly, in spite of both MTA1 and METTL3 are the catalytic core subunits, they have significant different SAM-binding activity. In the absence of METTL14, METTL3 alone shows strong SAM-binding activity[34]. While MTA1 exhibits no detectable SAM binding activity, even in the presence of MTA9. Our structural and biochemical data reveal that MTA1 is reliant on its binding partner, the accessory factor p2, for SAM binding. The NTD domain of p2 directly stimulates the SAM binding activity of MTA1. Compared to METTL3, MTA1 shows a distinct SAM binding mode, which probably provides more mechanistic understanding for DNA N6-adenine methylation.

The MTA1c complex also shows significant structural and mechanical differences from the bacterial DNA adenine methyltransferases, such as M.TaqI, M.EcoP15I, Dam, CcrM, and CamA, whose structure and catalytic mechanism have been well studied[19–23]. First, these bacterial enzymes are active as monomers or homodimers, while the MTA1c complex functions as heterotetramer. Second, only one or two subunits of bacterial enzymes are mainly involved in DNA recognition during methylation, while all four subunits of MTA1c methyltransferase contribute significantly to DNA binding during methylation. Bacterial enzymes contain a conserved TRD domain that is critical for DNA binding (Supplementary Fig. 7e and Supplementary Fig. 10), and this domain is likely not identified in MTA1c. Finally, the SAM-binding activity of MTA1 requires stimulation by the subunit of p2. This interesting activity regulation mechanism may be unique to MTA1c methyltransferase.

## Methods

**Protein expression and purification**. For the purification of *Tetrahymena thermophila* MTA1c (TthMTA1c), full-length *TthMTA1*, *TthMTA9* and *Tthp1* genes were synthesized by Generay Biotech and cloned into a modified pET28a-SUMO expression vector with an N-terminal SUMO-tag followed by a ubiquitin-like protein 1 (Ulp1) protease cleavage site using ligation independent cloning. The *Tthp2* gene was cloned into a pET28a-His₆SUMO vector. All fusion proteins were overexpressed in *E. coli* Rosetta (DE3) (Novagen) cells that were induced with 0.5 mM isopropyl-1-thio-β-D-galactopyranoside (IPTG) at OD₆₀₀ = 0.8 for 14 h at 18 °C respectively. Harvested cells expressing His₆SUMO-Tthp2 and excess cells expressing TthMTA1, TthMTA9 and Tthp1 were mixed and co-lysed by sonication in buffer containing 20 mM Tris, pH 7.5, 1 M NaCl. After centrifugation, the supernatant was incubated with Ni Sepharose (GE Healthcare), and the bound protein was eluted with buffer containing 20 mM Tris, pH 7.5, 300 mM NaCl, 300 mM imidazole. Eluted TthMTA1c protein was digested with Ulp1 protease at 4 °C to remove His₆SUMO and SUMO tag, and then further purified on a Heparin HP column (GE Healthcare), eluting with buffer containing 20 mM Tris, pH 7.5, 1 M NaCl. The protein was further purified by SEC (Superdex 200 Increase 10/300, GE Healthcare) in a buffer containing 20 mM Tris, pH 7.5, 300 mM NaCl, 2 mM DTT and then concentrated to a concentration of 15 mg/ml. All TthMTA1c mutants, ternary complexes, binary complexes and individual proteins were expressed and purified as wild-type TthMTA1c. All *Paramecium tetraurelia* (Pte) MTA9 mutants, PteMTA9-TthMTA1 binary complex and Ptep2-TthMTA1 binary complex were purified with an identical protocol.

For the experiment of GST pull-down, *TthMTA1*, *TthMTA9*, *Tthp1* and *Tthp2* were cloned into a modified pET23a vector (Novagen) with an N-terminal His$_6$GST-tag. Proteins were overexpressed in *E. coli* Rosetta (DE3) (Novagen) cells that were induced with 0.2 mM isopropyl-1-thio-β-D-galactopyranoside (IPTG) at OD$_{600}$ = 0.8 for 14 h at 18 °C. Cells were collected and lysed by sonication in buffer containing 20 mM Tris, pH 7.5, 1 M NaCl. After centrifugation, the supernatant was incubated with Ni Sepharose (GE Healthcare), and the bound protein was eluted with buffer containing 20 mM Tris, pH 7.5, 300 mM NaCl, 300 mM imidazole. Eluted His$_6$GST-TthMTA1 was purified on a Hitrap SP HP column (GE Healthcare) and His$_6$GST-TthMTA9, His$_6$GST-Tthp1 and His$_6$GST-Tthp2 were purified on a Heparin HP column (GE Healthcare). All proteins were further purified by size-exclusion chromatograph (Superdex 200 Increase 10/300, GE Healthcare) in a buffer containing 20 mM Tris, pH 7.5, 300 mM NaCl, 2 mM DTT.

Full-length human *METTL3* gene was amplified by PCR from HCT116 cells cDNA and cloned into pFastBac$^{TM}$ HTB with an N-terminal His$_6$-tag followed by a tobacco etch virus (TEV) protease cleavage site. Human *METTL14* gene was cloned into pFastBac$^{TM}$ Dual. The METTL3-METTL14 complex was co-expressed in Sf9 insect cells using a Bac-to-Bac baculovirus expression system. Harvested insect cells were lysed by sonication in buffer containing 20 mM Tris, pH 7.5, 150 mM NaCl, 1 mM Tris (2-carboxyethyl) phosphine (TCEP). After centrifugation, the supernatant was incubated with Ni Sepharose (GE Healthcare), and the bound protein was eluted with buffer containing 300 mM imidazole. The eluent was digested by His$_6$-tagged TEV protease and the mixture was dialyzed against 1.5 L buffer containing 20 mM Tris, pH 7.5, 150 mM NaCl, at 4 °C for 3 h. The cleaved protein was passed through the nickel column once again to remove all His$_6$-tagged proteins and was further purified by a Hitrap Q HP column (GE Healthcare) then the protein was further purified by SEC (Superdex 200 Increase 10/300, GE Healthcare) with buffer containing 20 mM Tris, pH 7.5, 150 mM NaCl, and 1 mM TCEP. The individual METTL3 protein was expressed and purified by using the similar method above.

**Crystallization, data collection and structure determination.** Initial crystallization screens for all TthMTA1-Tthp1-Tthp2 ternary complex, TthMTA1-Tthp2 binary complex, TthMTA1-Ptep2 binary complex, TthMTA1-PteMTA9 binary complex and TthMTA1 protein were carried out at 16 °C with using high-throughput crystallization screening kits (Hampton Research, Molecular Dimensions and QIAGEN). Well-diffracting crystals were manual optimized using the hanging drop vapor diffusion method at 16 °C. Crystals were obtained by mixing 1 μl complex solution and 1 μl of reservoir solution. Crystals of TthMTA1-Tthp1-Tthp2 ternary complex were grown from 0.05 M HEPES, pH 7.0, and 16% PEG 4000. The TthMTA1-Tthp2 binary complex was crystallized in 0.1 M MES, pH 6.5, and 15% PEG 550 MME. The TthMTA1-Ptep2 binary complex was crystallized in 0.1 M HEPES, pH 6.8, 0.2 M NaCl, and 10% PEG 8000. The TthMTA1-PteMTA9 binary complex was crystallized in 0.1 M Tris-HCl, pH 7.5, 0.2 M LiCl, and 14% PEG 8000. The crystals of TthMTA1 were grown from 0.15 M potassium sodium tartrate, and 18% PEG 3350. All crystals soaked in cryoprotectants made from the mother liquors supplemented with 20% (v/v) ethylene glycol and flash frozen in liquid nitrogen.

The SAM-bound and SAH-bound crystals of TthMTA1-Tthp1-Tthp2 ternary complex were obtained by soaking the apo crystals with 1 mM SAM or SAH for 30 min at 16 °C. These crystals then soaked in cryoprotectants made from the mother liquors supplemented with 20% (v/v) ethylene glycol and flash frozen in liquid nitrogen. The SAM-bound and SAH-bound crystals of TthMTA1-Tthp2 binary complex were generated by soaking the apo crystals with 1 mM SAM or SAH for 60 min at 16 °C. The SAM-bound crystals of TthMTA1-Ptep2 binary complex were generated by soaking the apo crystals with 1 mM SAM for 30 min at 16 °C. These crystals were cryoprotected using the corresponding reservoir solution, supplemented with 20% (v/v) glycerol, and flash-frozen in liquid nitrogen.

The X-ray diffraction data sets were collected at 100 K at beamline BL-17U1, BL-18U1 and BL-19U1 at Shanghai Synchrotron Radiation Facility (SSRF) in China. The diffraction data were indexed, integrated and scaled using the HKL3000 package[39]. The phases of TthMTA1-Tthp1-Tthp2 ternary complex and TthMTA1-Tthp2 binary complex were solved by Se single wavelength anomalous dispersion method using PHENIX AutoSol[40]. The SAM-bound and SAH-bound complexes structures were determined by molecular replacement using the structures of the ligand-free TthMTA1-Tthp1-Tthp2 and TthMTA1-Tthp2 complexes as the search model using the program PHASER[41]. The TthMTA1 structure was determined by molecular replacement using the structure of the TthMTA1 in TthMTA1-Tthp2 complexes as the search model. We used SWISS-MODEL to generate a homology model for the PteMTA9 MTase domain using the METTL14 MTase domain structure as a template. The TthMTA1-PteMTA9 complex structure was determined by molecular replacement using the structure of the TthMTA1 and the model structure of PteMTA9 as the search model. The atomic model was built manually in COOT[42]. Iterative cycles of crystallographic refinement were performed using PHENIX. All data processing and structure refinement statistics are summarized in Supplementary Tables 1,2,3. Structure figures were prepared using PyMOL (http://www.pymol.org/).

**Antibody-based methyltransferase activity assays.** Genomic DNA was extracted from overnight cultured vegetative *Thetrahymena* cells using TIANamp Genomic DNA Kit. The 954-bp dsDNA used in all methyltransferase assays was amplified by PCR from *Tetrahymena thermophila* strain SB210 genomic DNA using primers metGATC_F2 (5′-GTGCTATGCATTTTAAATTTATTCGCATTG AAGA-3′) and metGATC_R2 (5′-ATTCAGAATTTTAGTGTGTGGAGTATGAT AGTA-3′) as described previously and purified using a SteadyPure PCR DNA Purification Kit (Accurate Biology)[5].

For assays, a reaction cocktail containing 20 mM Tris, pH 7.5, 200 mM NaCl, 6 mM EDTA, 5 μM enzyme was prepared and 2.7 μg of substrate dsDNA was added in a 0.2 ml PCR tube, then 160 μM SAM was added to initiate the reaction with a final assay volume of 50 μl. The reaction was carried out for 3 h at 37 °C, then purified using a MinElute Reaction Cleanup Kit (QIAGEN). The purified dsDNA was denatured at 95 °C for 5 min, and kept on ice for 1 min. Afterwards, the samples of each group were spotted on a Hybond-N$^+$ membrane (GE Healthcare) using a Bio-Dot Apparatus (Bio-Rad), dried 5 min in air and further fixed to the membrane by exposed to the 254 nm UV light for 7 min. After blocking using 5% milk in TBST, the membrane was incubated with 1:1,000 anti-N6-methyladenosine antibody (Synaptic Systems, Cat #: 202003) at 4 °C overnight, then 1:2,000 Goat Anti-rabbit IgG/HRP antibody (Bioss, Cat #: bs-0295G-HRP) at room temperature for 1 h. The amount of input DNA was determined by Methylene blue staining. Each experiment was repeated three times.

**SAM-dependent methyltransferase activity assays.** SAM-dependent methyltransferase activity assay was displayed by using the MTase-Glo$^{TM}$ assay as described previously[19,43,44]. Briefly, an assay cocktail was prepared with 20 mM Tris, pH 7.5, 200 mM NaCl, 6 mM EDTA, 5 μM enzyme and 2.7 μg of 954-bp dsDNA substrate (or 50 μM 59 bp oligo dsDNA:5′-AACTTCTGTCATTA-CATTAAGCTTTAAAAAAAATTCAATTCTTTTATTTATTAGAATTATG-3′) was added in a 0.2 ml PCR tube, then 160 μM SAM was added to initiate the reaction with a final assay volume of 50 μl for 4 h at 37 °C. Reaction was quenched by adding trifluoroacetic acid (TFA) to 0.1% final concentration. Aliquots (8 μl) of the reaction were added to a 384-well black microplate (Corning), and mixed with 2 μl of 5× MTase-Glo$^{TM}$ reagent for 30 min at room temperature, and then 10 μl of 5× MTase-Glo$^{TM}$ detection solution were added to all wells and allowed to react in darkness for 30 min prior to reading. The luminescence signal was monitored by using the Synergy HTX multi-mode microplate reader (BioTek).

**GST pull-down assays.** 0.2 mg GST-tagged protein and 0.4 mg untagged TthMTA1, TthMTA9, PteMTA9, Tthp1 or Tthp2 protein were mixed at 4 °C in binding buffer containing 20 mM Tris, pH 7.5, 300 mM NaCl, 2 mM DTT. The mixed samples were incubated with 80 μl GST affinity resin at 4 °C for 60 min. The resin was washed five times in 1 ml binding buffer, then the proteins were eluted with 50 μl MBP buffer containing 20 mM Tris, pH 7.5, 300 mM NaCl, 2 mM DTT and 20 mM GSH. Elution samples were analyzed using 15% SDS-polyacrylamide gel electrophoresis (SDS–PAGE) and Coomassie blue staining. The assays were quantified by band densitometry. The experiment was repeated three times.

**Size-exclusion chromatography assays.** Size-exclusion chromatography assays were performed using a Superdex 200 Increase 10/300 gel filtration column (GE Healthcare) at a flow rate of 0.5 ml/min with absorbance monitored at 280 nm. For the TthMTA1 and TthMTA9 binding assays, the purified TthMTA1 and TthMTA9 were mixed at the molar ratio of 1:1 and incubated on ice for 60 min. For the TthMTA1, Tthp1 and Tthp2 binding assays, the purified TthMTA1, Tthp1 and Tthp2 were mixed at the molar ratio of 1:1:1 and incubated on ice for 60 min. For the TthMTA1, TthMTA9, Tthp1 and Tthp2 binding assays, the TthMTA1, TthMTA9, Tthp1 and Tthp2 were mixed at the molar ratio of 1:1:1:1 and incubated on ice for 60 min. The column was first equilibrated with the binding buffer containing 20 mM Tris, pH 7.5, 300 mM NaCl, 2 mM DTT, then 500 μl of sample was applied to the column. Fractions were analyzed on a 15% SDS-PAGE gel and visualized by Coomassie blue staining.

**Isothermal titration calorimetry (ITC) assays.** ITC assays were carried out on a MicroCal iTC200 isothermal titration calorimeter (Malvern) at 25 °C. All protein samples and SAM were prepared in the reaction buffer containing 20 mM Tris, pH 7.5, 300 mM NaCl. ITC assay for the binding of SAM to the MTA1c quaternary complex, 100 μl of SAM at 400 μM was titrated into 400 μl of MTA1c complex at 20 μM. For weak complexes, the measurement was repeated with increased concentrations. 800 μM SAM was titrated into 40 μM MTA1-p2, MTA1-MTA9-p2 and MTA1-p1-p2 complexes, 1200 μM SAM for 30 μM MTA1, MTA9, MAT1-MAT9, MAT1-p1, MTA9-p2, MTA1-MTA9-p1 titration. The first injection (1 μl) was followed by 19 injections of 2 μl. The stirring rate was 1000 r.p.m. The heat of dilution for SAM was measured for background subtraction. The titration curves were analyzed with OriginPro 2016 (MicroCal) with ones-site binding model.

**Electrophoretic mobility shift assays (EMSA).** Electrophoretic mobility shift assays (Fig. 7a) were carried out using a 27-bp dsDNA (5′-AACTTCTGTCAT-TACATTAAGCTTTAA-3′) purchased from Sangon Biotech. 4 μM dsDNA was incubated with increasing concentrations of binary, ternary or quaternary complexes of MTA1c in a buffer containing 20 mM Tris, pH 7.5, 300 mM NaCl in a total volume of 10 μl. The binding reactions were performed at 4 °C for 30 min,

then the reaction samples were analyzed on an 8% native polyacrylamide gels with Tris-Glycine buffer pH 8.5 and run at 100 volts for 40 min at 4 °C. Gels were visualized by using a ChemiDoc XRS + (Bio-rad).

**Statistics and reproducibility**. Antibody-based methyltransferase activity assays, SAM-dependent methyltransferase activity assays, GST pull-down assays, SEC assays and EMSA were repeated at least three times ($n \geq 3$) with similar results. Isothermal titration calorimetry assays were carried out at least twice ($n \geq 2$) with similar results.

**Reporting summary**. Further information on research design is available in the Nature Research Reporting Summary linked to this article.

## Data availability

The data supporting the findings of this study are available from the corresponding authors upon reasonable request. The atomic coordinates included in this study have been deposited in the Protein Data Bank (PDB) with the following accession codes: 7F4L, 7F4M, 7F4N, 7F4O, 7F4P, 7F4Q, 7F4R, 7F4S, and 7F4T. Source data for the figures and supplementary figures are provided as a Source Data file. Source data are provided with this paper.

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

## Acknowledgements

We are grateful to the staff of the BL-17U1, BL-18U1 and BL-19U1 beamlines at the National Center for Protein Sciences Shanghai (NCPSS) at Shanghai Synchrotron Radiation Facility (SSRF). This work was supported by grants from the National Natural Science Foundation of China (32022047), the Young Elite Scientists Sponsorship Program by CAST.

## Author contributions

J.C., R.H., H.C., and C.Y. expressed and purified the proteins and grew crystals. J.C., X.L., Y.C., W.X., and L.L. collected X-ray diffraction data. L.L. solved the structures. X.L., Y.C., R.H., and J.C. carried out all of the cloning and performed the biochemical assays. L.L. prepared the figures; L.L. wrote the paper and supervised all of the research.

## Competing interests

The authors declare no competing interests.
