## [Peer Review File · Nature Communications]

Title: Structural basis for MTA1c-mediated DNA N6-adenine methylationREVIEWER COMMENTS

Reviewer #1 (Remarks to the Author):

The epigenetic and epitranscriptomic modification of DNA and RNA respectively, has attracted a lot of attention in recent years. While N6-methyladenosine in RNA has been studied intensively and various roles of this modification in coding and non-coding RNAs have been described, the function of N6-adenine methylation in DNA (6mA) is less well understood. It was recently shown that the MTA1 complex, consisting of MTA1, MTA9, p1 and p2, can install 6mA in ciliates and that this modification can influence nucleosome occupancy in its vicinity and is required for cell cycle progression. Here, Chen and colleagues analyse the structure of this 6mA writer complex and the contribution of the different components to its catalytic activity. They find that MTA1 is the core subunit, which not only harbours the catalytic activity but also constitutes the assembly platform for the other proteins in the complex. More specifically, the MTA1-p2, MTA1-p1-p2 and MTA1-MTA9 complex are analysed at approx. 3Å resolution. This is complemented by mutants that affect interaction between the complex components, binding of SAM or DNA, and by enzymatic assays.

This study is well performed and, besides revealing details of the interactions and dynamics of this 6mA writer complex, the data are interesting in comparison with the structures and mechanisms of other methyltransferases, such as the METTL3/14 complex. The reviewer thinks that this manuscript could become a valuable contribution to Nature Communications if the following points are addressed:

Major points:

1. The authors suggest that the complex provides a channel for dsDNA binding and that MTA1, p1 and p2 contribute to its formation. They show that specific amino acid exchanges in these proteins lead to a reduction in MTase activity (Fig. 7e) and speculate that this could be due to reduced substrate binding. This needs to be consolidated, e.g., by EMSA or anisotropy experiments to actually reveal the effects on DNA substrate binding.
2. A key aspect of the manuscript, which makes it more interesting to a broad readership, is the comparison between the assembly, structure and function of MTA1c and other MTase complexes. This topic should be significantly expanded in the introduction and discussion, especially as the introduction is very short. Mentioning what is known about the structure and function of other multimeric MTases will make the manuscript more accessible and relevant, e.g. also to members of the fast-growing epitranscriptome community.
3. Does the complex also methylate RNA? Here the authors could use e.g. total RNA isolated from Tetrahymena, which seems to be available to them, or in vitro transcribe the PCR products they actually use in their assay for this manuscript. Some MTases seem to be promiscuous for DNA/RNA at least in vitro, which might also be the case for MTA1c.

Reviewer #2 (Remarks to the Author):

The manuscript titled “Structural basis for MTA1c-mediated DNA N6-adenine methylation” by Chen et al. provides structural and biochemical data to probe the mechanism of how the MTA1c complex fulfills methylation of DNA substrates. The authors provide evidence for the central role that MTA1 plays in binding three other factors, p1, p2, and MTA9. In addition, the authors also show that the catalytic site of MTA1 is critical for the in vitro methylation activity. The model that the authors present provides helpful architectural information about the MTA1c complex, shedding light into an interesting mechanistic model. However, some of the mechanistic models claimed by the authors need more direct evidence.

1. The authors exclude MTA9 as a SAM binding factor. However, even for MTA1, they could not detect SAM binding until p2 was added. Could MTA9 become active when another cofactor is added to the complex? Related to this, sequence variance in the SAM binding sites is not evidence for the lack of SAM-binding activity by MTA9 (p.14). SAM binding sites can vary as shown in many different structures. The authors also show that SAM can fit into the MTA9 catalytic site (Fig 6f). This again goes against what they discuss in P.14 (middle paragraph).
2. For ITC, the affinity for SAM is ~10 times greater when comparing MTA1-p1-p2 vs. MTA1-p1-p2-MTA9. How do the authors explain this?
3. What happens to SAM binding when the “catalytic” residues are mutated in MTA9? What happens to methylation (in vitro or in vivo) when the potential catalytic residue of MTA9 is mutated?
4. On page 10 line 246, lack of ligand density does not “confirm” that a factor cannot be catalytic.
5. The existence of an “autoinhibitory form” is not supported by the provided data. The authors postulate that rearrangement of the loops and side-chains near SAM might affect SAM binding. While it is reasonable to suggest that SAM binding might not be as favorable in certain complex states, observing minor conformational changes of the loops and side-chains does not warrant an “auto-inhibition”. Can the authors relieve the autoinhibition with a loss-of-function mutation? If not, induced-fit or stabilization of the SAM-bound state by cofactors would be more appropriate to describe the mechanism.
6. The model for DNA binding presented in Figure 7 needs to be clarified. In 7b, how the experimental data were used to extrapolate the model needs to be clearer, including which parts were modeled rather than empirically determined. The authors also presume that the binding site of the DNA/RNA substrates is known rather than proposed, for both METTL3 and MTA1. The proposed model is intriguing but needs clarity so that the reader is not confused about what is known and what is proposed.
7. The authors present the MTA9 EMSA data to suggest some experimental support for its role in binding DNA. If so, a mutational analysis of MTA9 to show how changing the basic patches of MTA9 affects EMSA and in vitro methylation (Fig 7e and f) would provide better support for the model they propose.
8. The authors emphasize the role of the DBD in DNA binding (Fig 7c), but even after mutating 3 different residues of DBD, the activity seems almost unaffected in Fig 7e. How do the authors reconcile this with their model?
9. The authors present an interesting model where multiple factors are involved in controlling SAM and DNA binding. How are they related? Does SAM binding affect the DNA affinity or vice versa?

10. At this modest resolution, there should not be any Ramachandran outliers.

11. For the MTA1-p1-p2-SAM structure, the gap between R_{work} and R_{free} is high. There may be overfitting of data.

Reviewer #3 (Remarks to the Author):

In this manuscript, the authors determined a number of X-ray structures of DNA adenine methyltransferase (MTase) complexes in various configurations. The manuscript has potential, but suffers several serious defects in significance, experimental design and interpretation, detailed below. I will only focus on the major issues.

Significance. The target of the study, a 4-membered MTase complex was initially discovered in the ciliate *Oxytricha*. In contrast to metazoa, DNA adenine methylation is abundant in various unicellular eukaryotes. Thus, the statement is inaccurate on p3-4, “the catalytic mechanism and the critical enzyme on DNA 6mA modifications are poorly understood”. It is only true that the existence of adenine methylation in mammalian DNA remains a controversial topic. However, the manuscript addresses neither the existence of the mammalian enzyme(s) nor the catalytic mechanism responsible for generating 6mA in mammalian DNA. The authors have to be very careful to lay out the questions they attempted to address. Also, the term, “6mA deposition” is misused for an in vitro enzyme characterization or otherwise has to be defined in the first place.

The DNA adenine methylation and its catalytic mechanism has been well studied in the bacterial enzymes M.TaqI, M.EcoP15I, Dam, CcrM, and CamA (three of them were published in Nature Communications). All of these structures have double-stranded DNA bound forms, and two of them are hemodimers (M.EcoP15I and CcrM), which belong to the same subfamily of MTA1-MTA9 (the subject of the current manuscript). Thus, it is odd that the manuscript does not perform comparative studies with the well characterized DNA adenine MTases; instead, the authors compared their structures to human MettL3-14, a less well characterized enzyme without a bound RNA substrate (actually it is of an inactive form of the enzyme, see below). The major defect of the current manuscript is the lack of a DNA bound structure, which is essential for understanding of the catalytic mechanism as well as the active complex formation. The current manuscript does not reach to the same level of those already published, particularly in comparison with M.EcoP15I, which like MTA1-MTA9-p1-p2, forms a multi-subunit complex.

Experimental Design. The authors reported a total of 9 structures. These structures are of truncations as well as mismatches of proteins from different organisms (e.g., TthMTA1-PteMTA9). It is awfully difficult to follow why certain combinations were used, and particularly, whether each structure represents a step of complex assembly and/or a step along the reaction pathway. Without such relevant information, the texts related to Figures 3-6 are very tedious to read.

Interpretation. While the enzymatic activities were performed in full-length proteins, the structures were formed by various truncations. Are these various truncations used for structural determination active enzymes? This is important, because the crystallized MettL3-14 fragments were not active, thus the comparison with an inactive enzyme structure will be misleading.

In addition, the in vitro activity was detected by an antibody, which has limited linear range. Could the conventional SAM-dependent enzyme assay be performed? Could the assay use a synthetic oligonucleotide, in addition to genomic DNA?

We have addressed all comments point by point (in blue) and incorporated all changes in the revised version of our manuscript.

Reviewers' Comments (in black) and response (in blue).

Reviewer #1:

The epigenetic and epitranscriptomic modification of DNA and RNA respectively, has attracted a lot of attention in recent years. While N6-methyladenosine in RNA has been studied intensively and various roles of this modification in coding and non-coding RNAs have been described, the function of N6-adenine methylation in DNA (6mA) is less well understood. It was recently shown that the MTA1 complex, consisting of MTA1, MTA9, p1 and p2, can install 6mA in ciliates and that this modification can influence nucleosome occupancy in its vicinity and is required for cell cycle progression. Here, Chen and colleagues analyse the structure of this 6mA writer complex and the contribution of the different components to its catalytic activity. They find that MTA1 is the core subunit, which not only harbours the catalytic activity but also constitutes the assembly platform for the other proteins in the complex. More specifically, the MTA1-p2, MTA1-p1-p2 and MTA1-MTA9 complex are analysed at approx. 3Å resolution. This is complemented by mutants that affect interaction between the complex components, binding of SAM or DNA, and by enzymatic assays.

This study is well performed and, besides revealing details of the interactions and dynamics of this 6mA writer complex, the data are interesting in comparison with the structures and mechanisms of other methyltransferases, such as the METTL3/14 complex. The reviewer thinks that this manuscript could become a valuable contribution to Nature Communications if the following points are addressed:

We thank this reviewer for these encouraging comments, which helped us with greatly improving our manuscript.

Major points:

1. The authors suggest that the complex provides a channel for dsDNA binding and that MTA1, p1 and p2 contribute to its formation. They show that specific amino acid exchanges in these proteins lead to a reduction in MTase activity (Fig. 7e) and speculate that this could be due to reduced substrate binding. This needs to be consolidated, e.g., by EMSA or anisotropy experiments to actually reveal the effects on DNA substrate binding.

We thank the reviewer for this suggestion. To test whether the specific amino acid exchanges in MTA1, p1 and p2 proteins affect the binding of DNA substrate, we performed EMSA experiments using mutated MTA1, p1 and p2 proteins. Our results show that substrate binding is reduced by specific amino acid exchanges in MTA1, p1, or p2 (Supplementary Fig. 8). These data further confirm that these specific positively charged residues in the channel are essential for binding substrate.

2. A key aspect of the manuscript, which makes it more interesting to a broad readership, is the comparison between the assembly, structure and function of MTA1c and other MTase complexes. This topic should be significantly expanded in the introduction and discussion, especially as the introduction is very short. Mentioning what is known about the structure and function of other multimeric MTases will make the manuscript more accessible and relevant, e.g. also to members of the fast-growing epitranscriptome community.

We thank the reviewer for this great suggestion. We have expanded this topic in the introduction and discussion by adding comparisons of the assembly, structure and function of MTA1c with other MTase complexes.

3. Does the complex also methylate RNA? Here the authors could use e.g. total RNA isolated from *Tetrahymena*, which seems to be available to them, or in vitro transcribe the PCR products they actually use in their assay for this manuscript. Some MTases seem to be promiscuous for DNA/RNA at least in vitro, which might also be the case for MTA1c.

The MTA1c complex may not methylate RNA, or it may have very low RNA methylation activity. We performed methyltransferase activity assays using an RNA substrate from in vitro transcription the PCR products and an DNA substrate, respectively. The results show that the MTA1c complex methylates DNA but not RNA of the same sequence (Supplementary Fig. 1f).

Reviewer #2:

The manuscript titled “Structural basis for MTA1c-mediated DNA N6-adenine methylation” by Chen et al. provides structural and biochemical data to probe the mechanism of how the MTA1c complex fulfills methylation of DNA substrates. The authors provide evidence for the central role that MTA1 plays in binding three other factors, p1, p2, and MTA9. In addition, the authors also show that the catalytic site of MTA1 is critical for the in vitro methylation activity. The model that the authors present provides helpful architectural information about the MTA1c complex, shedding light into an interesting mechanistic model. However, some of the mechanistic models claimed by the authors need more direct evidence.

We thank the reviewer for the positive comments.

1. The authors exclude MTA9 as a SAM binding factor. However, even for MTA1, they could not detect SAM binding until p2 was added. Could MTA9 become active when another cofactor is added to the complex? Related to this, sequence variance in the SAM binding sites is not evidence for the lack of SAM-binding activity by MTA9 (p.14). SAM binding sites can vary as shown in many different structures. The authors

also show that SAM can fit into the MTA9 catalytic site (Fig 6f). This again goes against what they discuss in P.14 (middle paragraph).

We thank the reviewer for pointing out this issue. Although MTA9 has no detectable SAM-binding affinity in the presence of p2, p1, p1-p2, p2-MTA1 or p1-MTA1, respectively (Fig. R1 and Fig. 5a), we cannot exclude the possibility that MTA9 possesses SAM-binding activity after forming four-subunits complex with p1, p2 and MTA1. Additionally, we only studied the effects of two accessory factors (p1 and p2) on the activities of MTA1 and MTA9, and it is unknown whether there are other components that can regulate the activity of MTA1 and MTA9. Therefore, we cannot exclude the possibility that MTA9 possesses SAM-binding activity in the presence of other additional regulatory factors. We have re-wrote the relevant sentences or discussion in the revised manuscript.

Fig. R1: ITC assay measuring the SAM-binding affinity of TthMTA9 in the presence of Tthp1 or Tthp1-Tthp2.

2. For ITC, the affinity for SAM is ~10 times greater when comparing MTA1-p1-p2 vs. MTA1-p1-p2-MTA9. How do the authors explain this?

We thank the reviewer for pointing out this issue. As discussed above, MTA9 may have weak SAM-binding affinity after forming four-subunits complex with p1, p2 and MTA1, So MTA1-p1-p2-MTA9 (MTA1c complex) shows a higher SAM-binding affinity than MTA1-p1-p2. In addition, the binding of MTA9 and MTA1 can stabilize the conformation of MTA1c complex, possibly improving its SAM-binding affinity.

3. What happens to SAM binding when the “catalytic” residues are mutated in MTA9? What happens to methylation (in vitro or in vivo) when the potential catalytic residue of MTA9 is mutated?

To test whether the mutation of potential catalytic residue (N150) in MTA9 affects SAM binding and DNA methylation, we performed ITC experiments and in vitro methyltransferase activity assays using mutated MTA9 protein (N150A). Our results show that the alanine substitution of N150 shows little reduced SAM-binding affinity and DNA methylation activity (Supplementary Fig. 9a and 9b).

4. On page 10 line 246, lack of ligand density does not “confirm” that a factor cannot be catalytic.

We thank the reviewer for pointing out this issue. We have re-wrote this sentence in the revised manuscript.

5. The existence of an “autoinhibitory form” is not supported by the provided data. The authors postulate that rearrangement of the loops and side-chains near SAM might affect SAM binding. While it is reasonable to suggest that SAM binding might not be as favorable in certain complex states, observing minor conformational changes of the loops and side-chains does not warrant an “auto-inhibition”. Can the authors relieve the autoinhibition with a loss-of-function mutation? If not, induced-fit or stabilization of the SAM-bound state by cofactors would be more appropriate to describe the mechanism.

We thank the reviewer for pointing out this issue and giving great suggestions. We have re-wrote this part and changed the statement in our revised manuscript.

6. The model for DNA binding presented in Figure 7 needs to be clarified. In 7b, how the experimental data were used to extrapolate the model needs to be clearer, including which parts were modeled rather than empirically determined. The authors also presume that the binding site of the DNA/RNA substrates is known rather than proposed, for both METTL3 and MTA1. The proposed model is intriguing but needs clarity so that the reader is not confused about what is known and what is proposed.

We thank the reviewer for pointing out this issue. We have clearly clarified the proposed model for DNA binding presented in Figure 7.

7. The authors present the MTA9 EMSA data to suggest some experimental support for its role in binding DNA. If so, a mutational analysis of MTA9 to show how changing the basic patches of MTA9 affects EMSA and in vitro methylation (Fig 7e and f) would provide better support for the model they propose.

We thank the reviewer for this suggestion. We first mutated multiple positively charged residues of MTA9 and performed EMSA assays. The mutation of the three positively charged residues (R231A/K232A/K278A) far from the proposed DNA binding channel maintains the strong DNA binding ability, while the alanine substitution of the three positive charged residues (R243A/H244A/R246A) near the

proposed DNA binding channel reduces the DNA binding affinity (Fig. R2a and R2b). We then performed in vitro methylation assay with MTA9 mutant. R243A/H244A/R246A shows reduced DNA methylation activity, suggesting that residues R243, H244 and R246 play an important role in binding DNA (Fig. R2c). These results are included in the revised manuscript (Fig. 7e and 7f, and Supplementary Fig. 8).

Fig. R2: Substrate DNA binding by MTA9. **a** The surface electrostatic potential calculated with PyMOL of MTA1c complex. The positively charged residues of MTA9 are indicated by orange arrows. **b** Electrophoretic mobility shift assay of the MTA1c complexes with indicated MTA9 point mutations. **c** In vitro methyltransferase activity of the MTA1c complexes with indicated MTA9 point mutations.

8. The authors emphasize the role of the DBD in DNA binding (Fig 7c), but even after mutating 3 different residues of DBD, the activity seems almost unaffected in Fig 7e. How do the authors reconcile this with their model?

We thank the reviewer for pointing out this issue. Our EMSA assays show that the four subunits of MTA1c complex collectively contribute to DNA binding (Fig. 7a). Additionally, mutations of multiple positively charged residues in the proposed DNA binding channel reduce DNA binding (Supplementary Fig. 8). These results suggest that there are many amino acids involved in binding DNA, and some residues may

have weak affinity for DNA binding. Therefore, mutations of a few of these residues may have little effect on DNA binding and enzyme activity.

9. The authors present an interesting model where multiple factors are involved in controlling SAM and DNA binding. How are they related? Does SAM binding affect the DNA affinity or vice versa?

We thank the reviewer for asking an interesting question. We're actually trying to figure that out. To test whether SAM binding affect the DNA affinity or vice versa, we performed EMSA and ITC assays, respectively (Supplementary Fig. 7a and 7b). Our results show that SAM binding does not improve DNA affinity (Supplementary Fig. 7a) and vice versa (Supplementary Fig. 7b). This indicates that SAM binding may have no significant effect on DNA affinity, and DNA binding may also have no significant effect on SAM affinity.

10. At this modest resolution, there should not be any Ramachandran outliers.

We thank the reviewer for pointing out this issue. We have re-refined our structures. There are no Ramachandran outliers to report. See the revised Supplementary Table. 1, Supplementary Table. 2 and Supplementary Table. 3.

11. For the MTA1-p1-p2-SAM structure, the gap between Rwork and Rfree is high. There may be overfitting of data.

We thank the reviewer for pointing out this issue. We have re-refined the MTA1-p1-p2-SAM complex structure. Now the gap between Rwork and Rfree is modest (Supplementary Table. 1, Rwork: 0.2507, Rfree: 0.2758).

Reviewer #3:

In this manuscript, the authors determined a number of X-ray structures of DNA adenine methyltransferase (MTase) complexes in various configurations. The manuscript has potential, but suffers several serious defects in significance, experimental design and interpretation, detailed below. I will only focus on the major issues.

We thank this reviewer for the comments, which helped us with greatly improving our manuscript.

Significance. The target of the study, a 4-membered MTase complex was initially discovered in the ciliate *Oxytricha*. In contrast to metazoa, DNA adenine methylation is abundant in various unicellular eukaryotes. Thus, the statement is inaccurate on p3-4, "the catalytic mechanism and the critical enzyme on DNA 6mA modifications

are poorly understood”. It is only true that the existence of adenine methylation in mammalian DNA remains a controversial topic. However, the manuscript addresses neither the existence of the mammalian enzyme(s) nor the catalytic mechanism responsible for generating 6mA in mammalian DNA. The authors have to be very careful to lay out the questions they attempted to address. Also, the term, “6mA deposition” is misused for an in vitro enzyme characterization or otherwise has to be defined in the first place.

We thank the reviewer for pointing out this issue. We have corrected these statement and term.

The DNA adenine methylation and its catalytic mechanism has been well studied in the bacterial enzymes M.TaqI, M.EcoP15I, Dam, CcrM, and CamA (three of them were published in Nature Communications). All of these structures have double-stranded DNA bound forms, and two of them are hemodimers (M.EcoP15I and CcrM), which belong to the same subfamily of MTA1-MTA9 (the subject of the current manuscript). Thus, it is odd that the manuscript does not perform comparative studies with the well characterized DNA adenine MTases; instead, the authors compared their structures to human MettL3-14, a less well characterized enzyme without a bound RNA substrate (actually it is of an inactive form of the enzyme, see below).

We thank the reviewer for pointing out this issue. We have added the comparative studies between the MTA1c complex and the bacterial enzymes M.TaqI, M.EcoP15I, and CcrM in our revised manuscript (Supplementary Fig. 7e and Supplementary Fig. 10).

The major defect of the current manuscript is the lack of a DNA bound structure, which is essential for understanding of the catalytic mechanism as well as the active complex formation. The current manuscript does not reach to the same level of those already published, particularly in comparison with M.EcoP15I, which like MTA1-MTA9-p1-p2, forms a multi-subunit complex.

We agree that determination of a DNA bound structure is essential for understanding of the catalytic mechanism as well as the active complex formation. Actually, we designed a lot of DNA, and tried to solve a DNA bound MTA1c complex structure. However, we did not obtain a stable MTA1c-DNA complex using these DNAs. Although we were unable to determine the structural basis of DNA recognition by MTA1c, our biochemical data provided important information for DNA binding, and these information, combined with our 9 structures, are very helpful in understanding the catalytic mechanism of MTA1c complex.

Although M.EcoP15I is a multi-subunit complex, our results suggest that the MTA1c complex is significantly different from M.EcoP15I in terms of structural assembly,

DNA recognition and cofactor binding. First, M.EcoP15I is a prototype of the Type III R–M family, consisting of two methylation (Mod) and one restriction (Res) subunits, where only two Mod subunits (homodimer) are responsible for DNA methylation, while the Res subunit is not. The Res subunit is responsible for ATP hydrolysis and cleavage. In contrast, The MTA1c methyltransferase is a four-subunit complex, all four of which are essential for DNA methylation. Second, only one Mod subunit of M.EcoP15I is mainly involved in DNA recognition during methylation, while all four subunits of MTA1c methyltransferase contribute significantly to DNA binding during methylation. Third, The SAM-binding activity of MTA1 requires stimulation by the subunit of p2, which is significantly different from that of M.EcoP15I and other methyltransferases. Together, our structural and biochemical data shed new light on the mechanisms of DNA methylation by methyltransferases.

Experimental Design. The authors reported a total of 9 structures. These structures are of truncations as well as mismatches of proteins from different organisms (e.g., TthMTA1-PteMTA9). It is awfully difficult to follow why certain combinations were used, and particularly, whether each structure represents a step of complex assembly and/or a step along the reaction pathway. Without such relevant information, the texts related to Figures 3-6 are very tedious to read.

We apologize for the un-clarity in these structural information mentioned above. We sought to solve the full-length protein structures by crystallization, but did not get any crystal after screening a wide range of conditions. Therefore, we removed some of the flexible regions of proteins and solved the structures of the truncated complexes. We were also unable to obtain suitable crystals using TthMTA1-TthMTA9 and PteMTA1-PteMTA9 complex proteins for crystallization, although we spent a lot of time and screened a large number of crystallization conditions. Interesting, we got good crystals for diffraction using the TthMTA1-PteMTA9 combination proteins. Due to the high sequence similarity between TthMTA9 and PteMTA9, PteMTA9 can replace TthMTA9 to form a stable and active MTA1c methyltransferases. Therefore, we determined the structural basis of the TthMTA1-PteMTA9 binary complex for understanding the mechanism of DNA adenine methylation. In addition, the determination of all structures that represent each step of complex assembly is also very difficult. We have determined only the structures that represent the key steps of complex assembly. And since there are four subunits (MTA1, MTA9, p1, p2) and two cofactors (SAM and SAH) contained in our nine structures, the cross combination of these multiple factors may complicate our structural information. For clarity, we have added relevant structure information for our structures in the revised manuscript.

Interpretation. While the enzymatic activities were performed in full-length proteins, the structures were formed by various truncations. Are these various truncations used

for structural determination active enzymes? This is important, because the crystallized MettL3-14 fragments were not active, thus the comparison with an inactive enzyme structure will be misleading.

We thank the reviewer for pointing out this issue. In order to test whether the truncations used for structural determination are active enzymes, we performed in vitro methyltransferase activity assays using various truncated proteins. Our result shows that these various truncations used for structural determination maintained enzyme activity similar to that of wild type (Supplementary Fig. 2h), indicating that these truncations are active enzymes.

In addition, the in vitro activity was detected by an antibody, which has limited linear range. Could the conventional SAM-dependent enzyme assay be performed? Could the assay use a synthetic oligonucleotide, in addition to genomic DNA?

According to the reviewer's suggestion, we performed the conventional SAM-dependent enzyme assay- Promega luminescence assay (Hsiao, K. et al., 2016; Dai, S. et al., 2020; Zhou, J. et al., 2021) using MTA1c proteins. Similar to the results of antibody detection assay, the MTA1c quaternary complex showed methylation activity for both genomic DNA and oligonucleotide, while the single MTA1 subunit and MTA1-MTA9 binary complex did not (Supplementary Fig. 1c-1e).

REVIEWER COMMENTS

Reviewer #1 (Remarks to the Author):

During the revision of their manuscript, the authors have improved the manuscript and performed several additional experiments and analyses that, in the opinion of this reviewer, address the key issues raised.

Reviewer #2 (Remarks to the Author):

The manuscript is improved. However, here are some things that still need to be addressed.

1. Fig 7a has too high contrast and important information might be missing.
2. Supp Fig 9 has a critical typo (Aln?)
3. Supp Fig 7d should remove the RNA/DNA cartoon. Some mutations were tested for the two methyltransferase complexes, but limited mutagenesis studies cannot deduce the path of the substrate RNA or DNA. Highlighting important patches is justified, but a diagram of a path is misleading because there is not enough experimental data to support it. To prevent overinterpretation, remove the yellow dashed lines.
4. Supp fig 1f: To do a better comparison, ssDNA should also be included. This would show whether the specificity is with secondary structure or the DNA backbone.
5. Fig 6c and 6d are redundant and should be combined. The figure legend also has a typo.
6. Fig 6a and Fig 6b are confusing because the authors are reporting a real structure of SAM in p2 and p1-p2 bound conformations. Why did they have to model SAM when they have structures?
7. Fig 7c should indicate that this is a model! Indicate clearly which ones have been imported from superimposition, so that it is not confusing which structures are from modeling and which ones are from empirical data.
8. There are many typos and grammatical errors. Please fix them for clarity.

Reviewer #3 (Remarks to the Author):

In this revised manuscript, unfortunately the authors made limited efforts to address my concerns. Thus, I cannot endorse the current manuscript for publication and have reiterated some of my points again.

Significance. In the title and/or abstract, the authors should state clearly that the MTA1c is from ciliate, and that proteins assembled from *Tetrahymena thermophila* and *Paramecium tetraurelia* were used in the study. The biology of DNA 6mA methylation in ciliate should be the focus of this paper. Instead, the

first paragraph of the Introduction and first 10+ references cite DNA 5-methylcytosine and have nothing to do with N6-methyladenine. Please refer to the latest publication on the debatable topic, PMID: 35113697. It is essential that the authors distinguish between 5-methylcytosine and N6-methyladenine -- the two types of DNA methylation should not be referenced without differentiation or specification.

Experimental Design. Focusing on ciliate does not diminish the significance of this paper. However, the logic from *Oxytricha* to *Tetrahymena* to *Paramecium* should be clearly laid out in this paper. The current manuscript leads you to believe that the success of crystallization by mismatched proteins from different organisms is the only success criteria, as the authors stated in the reply-to-reviewers “we removed some of the flexible regions of proteins and solved the structures of the truncated complexes,” but the information and justification of the approaches were not in the text. To provide some justification, for example, please address:

On p8, in the “Specific recognition of MTA9 by MTA1” section please explain why TthMTA1 has to recognize PteMTA9.

Similarly, two related structures were determined, (1) TthMTA1 and Tthp2 and (2) SAM-bound TthMTA1-Ptep2, please explain why SAM-bound was done with the mismatched complex of TthMTA1 and Tthp2.

Interpretation

The new data provided in the revision is in Supplementary figure 2h. Unfortunately, the new data didn't answer the questions but raised more concerns. No sufficient information provided: (1) what were the substrates used in the assay, (2) what was the method for the detection, and (3) what truncations were used in the assay and from which organisms?

From the data, the authors claimed broadly on p8 that “The truncations used for structural determination are active enzymes (Supplementary Fig. 2h)”. If this is true, is the “TthMTA1 MTase domain (residues 171-372)” active? It is essential to define what truncations were used in this particular experiment (the same should apply to other data panels).

Figure 1 legend, genomic DNA (is it from *Tetrahymena thermophila*?) was used in the assay. *Tetrahymena* contains a significant amount of DNA 6mA (see PMID 35113697). Please explain why the antibody used in the assay does not detect the signal. What is the detection level of the antibody? What is the linear range of the antibody detection? How did the author convert the dot blot intensity to the relative activity?

The methods section stated that genomic DNA was used in all methyltransferase assays. Does this also include the data in Supplementary Fig. 2h?

A new method section on MTase-Glo assay (p24) was added. What data was derived from this assay?

Which experiment used 59-bp oligo dsDNA? How many potential target sites within the 59-bp? Did the reaction byproduct SAH, measured by MTase-Glo, have the same final molarity as the DNA substrate used in the reaction? Please provide [SAH] concentrations measured, instead of % of activity.

For the ITC experiments, why is the N number (molar ratio) changed in a wide range (~0.4 to ~1.2)? Please explain.

We are very grateful to the reviewers, who read the manuscript carefully and provided many thoughtful suggestions. We have addressed all comments point by point (in blue) and incorporated all changes in the revised version of our manuscript.

Reviewers' Comments (in black) and response (in blue).

Reviewer #1:

During the revision of their manuscript, the authors have improved the manuscript and performed several additional experiments and analyses that, in the opinion of this reviewer, address the key issues raised.

We thank this reviewer for these encouraging comments, which helped us with greatly improving our manuscript.

Reviewer #2:

The manuscript is improved. However, here are some things that still need to be addressed.

1. Fig 7a has too high contrast and important information might be missing.

We thank the reviewer for pointing out this issue. We have adjusted the contrast of Fig 7a in the revised manuscript.

2. Supp Fig 9 has a critical typo (Aln?)

We have corrected it.

3. Supp Fig 7d should remove the RNA/DNA cartoon. Some mutations were tested for the two methyltransferase complexes, but limited mutagenesis studies cannot deduce the path of the substrate RNA or DNA. Highlighting important patches is justified, but a diagram of a path is misleading because there is not enough experimental data to support it. To prevent overinterpretation, remove the yellow dashed lines.

We thank the reviewer for this suggestion. We have removed the RNA cartoon (yellow dashed lines) in Supp Fig 7d.

4. Supp fig 1f: To do a better comparison, ssDNA should also be included. This would show whether the specificity is with secondary structure or the DNA backbone.

We thank the reviewer for this great suggestion. We performed methyltransferase activity assays using an ssDNA substrate. The result shows that the MTA1c complex

has very low ssDNA methylation activity. This suggests that the specificity may depend on the secondary structure of dsDNA.

5. Fig 6c and 6d are redundant and should be combined. The figure legend also has a typo.

We thank the reviewer for this suggestion. In the revised version, Fig 6c and 6d were combined into the new Fig 6c.

The typo has been corrected in the revised version.

6. Fig 6a and Fig 6b are confusing because the authors are reporting a real structure of SAM in p2 and p1-p2 bound conformations. Why did they have to model SAM when they have structures?

We thank the reviewer for pointing out this issue.

We apologize for this confusing information. To determine the conformation differences of the SAM-binding pocket before SAM binding in different MTA1 structures, we compared the SAM-free MTA1 structures in the MTA1-MTA9, MTA1-p2 and MTA1-p1-p2 complexes (Fig 6a and 6b). For a clear comparison, a SAM molecular was modelled in the SAM binding-pocket of MTA1. Because the SAM-binding pocket also has a small conformational difference between the SAM-bound state and the SAM-free state, in order to avoid the conformational difference caused by the SAM binding, we show the model structure of SAM in p2, p1-p2 and MTA9 bound conformations.

We have revised the figure legend of Figure 6 for clarity.

7. Fig 7c should indicate that this is a model! Indicate clearly which ones have been imported from superimposition, so that it is not confusing which structures are from modeling and which ones are from empirical data.

We thank the reviewer for pointing out this issue. In the revised manuscript, we have clearly clarified the proposed model for DNA binding presented in Fig 7c.

8. There are many typos and grammatical errors. Please fix them for clarity.

We have read the manuscript carefully and corrected typos and grammatical errors.

Reviewer #3:

In this revised manuscript, unfortunately the authors made limited efforts to address my concerns. Thus, I cannot endorse the current manuscript for publication and have reiterated some of my points again.

Significance. In the title and/or abstract, the authors should state clearly that the MTA1c is from ciliate, and that proteins assembled from Tetrahymena thermophile

and *Paramecium tetraurelia* were used in the study. The biology of DNA 6mA methylation in ciliate should be the focus of this paper. Instead, the first paragraph of the Introduction and first 10+ references cite DNA 5-methylcytosine and have nothing to do with N6-methyladenine. Please refer to the latest publication on the debatable topic, PMID: 35113697. It is essential that the authors distinguish between 5-methylcytosine and N6-methyladenine -- the two types of DNA methylation should not be referenced without differentiation or specification.

We thank the reviewer for pointing out this issue.

In the revised abstract, we have stated clearly that the MTA1c is from ciliate, and that proteins assembled from *Tetrahymena thermophila* and *Paramecium tetraurelia* were used in the study.

We have revised the Introduction and removed most references about the functional studies of DNA 5-methylcytosine in the first paragraph of the Introduction. Now, the first paragraph of the Introduction mainly focuses on DNA 6mA methylation. We also cited the latest study on the identification and quantification of 6mA (PMID: 35113693).

Experimental Design. Focusing on ciliate does not diminish the significance of this paper. However, the logic from Oxytricha to *Tetrahymena* to *Paramecium* should be clearly laid out in this paper. The current manuscript leads you to believe that the success of crystallization by mismatched proteins from different organisms is the only success criteria, as the authors stated in the reply-to-reviewers “we removed some of the flexible regions of proteins and solved the structures of the truncated complexes,” but the information and justification of the approaches were not in the text. To provide some justification, for example, please address:

On p8, in the “Specific recognition of MTA9 by MTA1” section please explain why TthMTA1 has to recognize PteMTA9.

We thank the reviewer for pointing out this issue. Despite many attempts, the TthMTA9-TthMTA1 complex crystal was not obtained. Previous study has shown that 6mA is abundant in *Paramecium tetraurelia* (Pte) and *Tetrahymena thermophila* (6mA/A>0.1%). (PMID: 31722409). Sequence alignment result shows that MTA1c subunits are highly conserved between *Tetrahymena thermophila* and *Paramecium tetraurelia*. This implies that similar protein machines may be responsible for 6mA modification in both species. In fact, we have successfully purified PteMTA1c complex by recombinant expression. Therefore, we speculated that the interactions between the MTA1c subunit proteins in these two species were highly conserved. Structure prediction suggests that the rigid structures of MTA9 in *Tetrahymena thermophila* and *Paramecium tetraurelia* are remarkably similar. Both pull-down and co-purification assays indicate that TthMTA1 and PteMTA9 can form a stable complex (Fig. 2h). According to sequence alignment and structural information, PteMTA9 residues involved in TthMAT1 interaction are also conserved in TthMTA9. These results suggest that TthMTA1 can specifically recognize PteMTA9.

We have added the explanation in the “Structure of MTA1 in complex with MTA9” section (First paragraph)

Similarly, two related structures were determined, (1) TthMTA1 and Tthp2 and (2) SAM-bound TthMTA1-Ptep2, please explain why SAM-bound was done with the mismatched complex of TthMTA1 and Tthp2.

We thank the reviewer for pointing out this issue. It is an interesting finding that p2 can directly activate the SAM-binding activity of MTA1. To further confirm this finding and better understand the activation mechanism, we determined the SAM-bound TthMTA1-Tthp2 and SAM-bound TthMTA1-Ptep2 structures. Since Tthp2 and Ptep2 are conserved proteins, both can form stable complex with TthMTA1.

We have clarified in the “p2 directly activates the cofactor-binding activity of MTA1” section (Second paragraph)

Interpretation

The new data provided in the revision is in Supplementary figure 2h. Unfortunately, the new data didn't answer the questions but raised more concerns. No sufficient information provided: (1) what were the substrates used in the assay, (2) what was the method for the detection, and (3) what truncations were used in the assay and from which organisms?

We apologize for the un-clarity in this question. We have provided the sufficient information in the revised figure legend of Supplementary figure 2h.

We used a 954-bp DNA as the substrate in the assay. The 954-bp DNA was amplified by PCR from *Tetrahymena thermophila* strain SB210 genomic DNA. Antibody-based methyltransferase activity assay was used for the detection. The proteins used in the experiment were all from *Tetrahymena thermophila*. “ Δ ” indicated the MTA1c protein was truncated, specifically as follows: Δ TthMTA1 (residues 126-372), Δ TthMTA9 (residues 67-449) and Δ Tthp1 (residues 1-309).

From the data, the authors claimed broadly on p8 that “The truncations used for structural determination are active enzymes (Supplementary Fig. 2h)”. If this is true, is the “TthMTA1 MTase domain (residues 171-372)” active? It is essential to define what truncations were used in this particular experiment (the same should apply to other data panels).

We thank the reviewer for pointing out this issue. We have rewritten this sentence and provided sufficient truncation information in the revised figure legend of Supplementary figure 2h.

Figure 1 legend, genomic DNA (is it from *Tetrahymena thermophila*?) was used in the assay. *Tetrahymena* contains a significant amount of DNA 6mA (see PMID

35113697). Please explain why the antibody used in the assay does not detect the signal. What is the detection level of the antibody? What is the linear range of the antibody detection? How did the author convert the dot blot intensity to the relative activity?

We apologize for the inaccurate description. The substrate used in the assay was a 954-bp dsDNA amplified by PCR from *Tetrahymena thermophila* strain SB210 genomic DNA. It doesn't have a 6mA modification.

Dot blot intensity in the assay was quantitatively analyzed by the gray scanning. The intensity of MTA1c (wild-type)-catalyzed group was defined as 100%, and all intensity values have been normalized to the intensity value of MTA1c-catalyzed group.

To evaluate whether antibody-based methyltransferase activity assay can detect differences in enzyme activity, we generated a scatter chart by plotting dot blot intensity (Y-axis) against substrate (954-bp dsDNA) concentration (X-axis). In this experiment, wild-type TthMTA1c was used to catalyze the 6mA methylation of the 954-bp dsDNA as described in the method. The quantification reflects the relative amounts as a ratio of each 6mA DNA dot relative to the dot of TthMTA1c-catalyzed group. Though this method has limit linear range at high 6mA concentration, the intensity signal of 6mA is still related to the amount of substrate DNA.

We tried to detect the activity by using SAM-dependent enzyme activity assay, due to the low enzyme activity probably, we did not detect the activity of some mutants by SAM-dependent enzyme assay, but showed a 6mA signal by antibody-based method. Antibody-based methyltransferase activity assay has high sensitivity and is widely used to study and detect DNA 6mA modification (PMID: 32183942, PMID: 31104845, PMID: 30017583, PMID: 25936839, PMID: 27713410). Therefore, we used the antibody-based enzyme assay in our study.

The methods section stated that genomic DNA was used in all methyltransferase assays. Does this also include the data in Supplementary Fig. 2h?

The substrate used in the assay was a 954-bp dsDNA amplified by PCR from *Tetrahymena thermophila* strain SB210 genomic DNA. We have clarified it in the figure legend of Supplementary Fig. 2h.

A new method section on MTase-Glo assay (p24) was added. What data was derived from this assay? Which experiment used 59-bp oligo dsDNA? How many potential target sites within the 59-bp? Did the reaction byproduct SAH, measured by MTase-Glo, have the same final molarity as the DNA substrate used in the reaction? Please provide [SAH] concentrations measured, instead of % of activity.

The methyltransferase activity shown in Supplementary Fig. 1d and 1e was detected by the MTase-Glo assay. We clarified it in the figure legends. In Supplementary Fig. 1d, the 59-bp oligo dsDNA was used as the substrate to detect methyltransferase activity. The 59-bp dsDNA (5'-AACTTCTGTCATTACATTAAGCTTTAAAAATTCAATTCTTTATTTATTAG AATTATG-3') has sixteen potential target sites. 50 μ M 59-bp oligo dsDNA was used for the reaction, but the amount of the byproduct SAH was much less than the input DNA substrate. This indicates low reactivity, probably due to the weak interaction between MTA1c and oligo DNA. Supplementary Fig. 1d and 1e show the measured concentrations of [SAH].

For the ITC experiments, why is the N number (molar ratio) changed in a wide range (~0.4 to ~1.2)? Please explain.

We thank the reviewer for pointing out this issue. The N value in ITC is known to be the “binding stoichiometry”. The low N number in our ITC experiments was mainly reflected in MTA1-MTA9-p1 ($N=0.38\pm0.03$) and METTL3-METTL14 ($N=0.44\pm0.01$). We have re-purified the protein and experimented with different concentrations of the protein and conditions, and found that the value of N was still much less than 1. The low N value of MTA1-MTA9-p1 may be due to early saturation or only part of the protein is conformationally active, although the protein used in the experiment is tested for homogeneity by SEC. For METTL3-METTL14, some proteins may have bound to SAM when expressed in insect cells, resulting in fewer proteins that can actually bind to SAM in the ITC experiment.

REVIEWERS' COMMENTS

Reviewer #2 (Remarks to the Author):

I am mostly satisfied by the revisions that the authors have included. There is one unresolved issue.

For Fig 7c, the authors should put a clear label in the figure itself that it is a model. While this is described in the figure legend, the way this figure is designed (including the inset) it looks like this entire complex was determined by crystallography. Please include which segments originate from which structure or dataset, so that the reader can interpret which parts are put together through modeling. And clearly include "Model" on the figure itself.

Reviewer #3 (Remarks to the Author):

I have no further comments. However, it was a surprise to have so many experimental details (after multiple round of reviews) were missing based on the reply-to-reviewers comments.

We are very grateful to the reviewers, who read the manuscript carefully and provided many thoughtful suggestions. We have addressed all comments point by point (in blue) and incorporated all changes in the revised version of our manuscript.

Reviewers' Comments (in black) and response (in blue).

Reviewer #2:

I am mostly satisfied by the revisions that the authors have included. There is one unresolved issue.

For Fig 7c, the authors should put a clear label in the figure itself that it is a model. While this is described in the figure legend, the way this figure is designed (including the inset) it looks like this entire complex was determined by crystallography. Please include which segments originate from which structure or dataset, so that the reader can interpret which parts are put together through modeling. And clearly include “Model” on the figure itself.

We thank the reviewer for pointing out this issue. In the revised manuscript, we have put a clear label in the figure. Now, the Figure 7c not only clearly includes “Model” on the figure itself, but also indicates the segments origin of the MTA1c-DNA model structure.

Reviewer #3:

I have no further comments. However, it was a surprise to have so many experimental details (after multiple round of reviews) were missing based on the reply-to-reviewers comments.

We thank this reviewer for the thoughtful review of our manuscript. We have revised the Methods section and Reporting summary and provided sufficient details of the experiments. We also appreciate the constructive criticisms and suggestions, which helped us with greatly improving our manuscript. We have carefully taken these criticisms and suggestions into consideration in preparing our revision, which has resulted in a manuscript that is clearer, more compelling, and broader.